# ROUTING, CASCADES, AND USER CHOICE FOR LLMS

**Rafid Mahmood**
University of Ottawa & NVIDIA
mahmood@telfer.uottawa.ca

## ABSTRACT

To mitigate the trade-offs between performance and costs, LLM providers route user tasks to different models based on task difficulty and latency. We study the effect of LLM routing with respect to user behavior. We propose a game between an LLM provider with two models (standard and reasoning) and a user who can re-prompt or abandon tasks if the routed model cannot solve them. The user's goal is to maximize their utility minus the delay from using the model, while the provider minimizes the cost of servicing the user. We solve this Stackelberg game by fully characterizing the user best response and simplifying the provider problem. We observe that in nearly all cases, the optimal routing policy involves a static policy with no cascading that depends on the expected utility of the models to the user. Furthermore, we reveal a misalignment gap between the provider-optimal and user-preferred routes when the user's and provider's rankings of the models with respect to utility and cost differ. Finally, we demonstrate conditions for extreme misalignment where providers are incentivized to throttle the latency of the models to minimize their costs, consequently depressing user utility. The results yield simple threshold rules for single-provider, single-user interactions and clarify when routing, cascading, and throttling help or harm.

## 1 INTRODUCTION

Large language models (LLMs) have exploded into general-purpose technologies developed by a small set of model providers that permit users to perform diverse applications via a natural language interface (Maslej et al., 2025). Often users can subscribe to a provider, who affords users access to a family of models differentiated by quality and latency[1]. Providers face inference costs for each use that vary depending on the model and hardware (Leviathan et al., 2023). This presents a tiered ecosystem with a quality-latency-cost trade-off.

Providers navigate these trade-offs via LLM routing and cascading (Chen et al., 2023; Hu et al., 2024). Routing involves selecting an appropriate model per-use by reserving difficult tasks for larger, more expensive models and easier tasks for smaller, cheaper models. Cascading routes tasks over multiple rounds to smaller models first and escalating to larger ones if necessary. Routing has become industry practice, e.g., GPT-5 routes tasks between 'a smart, efficient model that answers most questions, [and] a deeper reasoning model for harder problems' (OpenAI, 2025).

While routing literature includes algorithms to estimate LLM performance and optimize the quality-latency-cost trade-off (Ding et al., 2024; Ong et al., 2025; Dekoninck et al., 2025), they often treat user response as exogenous. However, the prompt-based interface of LLMs implies a recurring cost for model failures if users repeatedly interact with the system (Naor, 1969; Garnett et al., 2002). Depending on the task, if a model fails to complete it, users may be patient and prompt the model again, or give up on the task (Castro et al., 2023; Wester et al., 2024; Krishna et al., 2024; Fu et al., 2025). In the worst case, user dissatisfaction may lead to un-subscription from the provider service. Consequently, optimizing for single-pass costs may be penny-wise but welfare-foolish.

In this paper, we study the welfare gap between a single LLM provider with two models (standard and reasoning) and a user under a multi-round LLM routing game (Simaan & Cruz Jr, 1973). Given

---

[1]For a list of LLM subscriptions, see: https://research.aimultiple.com/llm-pricing/#comparing-llm-subscription-plans.

**Only the reasoning model has high accuracy for its latency**

**Users** will abandon tasks if given to the standard model and not cascaded to the reasoning model.

Depending on the cost-of-pass of each model and the risk of unsubscribing, **providers** either route to the standard model, cascade at a minimum rate, or route directly to the reasoning model.

**Both models have high accuracies for their latencies**

**Users** are always patient and continue using regardless of the model choice.

**Providers** should route to the model with the lower cost-of-pass.

**Both models have low accuracy for their latencies**

**Users** are impatient & abandon tasks regardless of the model choice.

**Providers** should route to the cheaper model if the risk of users unsubscribing is low, otherwise route to the reasoning model.

**Only the standard model has high accuracy for its latency**

**Users** will abandon tasks if given to the reasoning model but will stay if given the standard model.

**Providers** should optimize over the user best response curve.

Figure 1: Key guidelines for optimal routing in the face of reactive users. User behavior depends on the state of the two models in terms of providing utility versus the delay incurred from inference compute. Provider policies must follow different thresholding rules in each region.

a task, the provider determines an initial model and a cascade rule on whether to escalate the task if the standard model fails. For each pass, the provider pays a compute cost and the user pays a latency cost (i.e., delay from model response). If the model fails to complete the task, the user may either re-prompt or churn the system, which imposes potential expected lost future revenue to the provider.

We prove the Stackelberg equilibrium, by first characterizing the user best response of when to churn versus re-prompt. We then show that the provider's problem reduces to a single-variable optimization problem that in many regimes yields simple threshold-based routing policies. Importantly, we show that cascades are rarely optimal outside specific regimes where the two models are sufficiently differentiated in value minus latency. Finally, we study the operational implications of simple routing heuristics, provider-user misalignment, and the effect of throttling latency. Figure 1 summarizes the key insights of expected user behavior and consequent routing policies. We observe:

- **User patience and sensitivity depend on the model net values.** When both models deliver positive (respectively, negative) value, users stay (respectively, churn) regardless of routing. When the models differ, users churn or stay depending on the cascade likelihood.

- **Optimal routing often collapses to simple threshold rules.** When models deliver similar value, the optimal policy is to route with no cascade to one of the models. In the mixed cases, some regimes reveal routing to the standard model and then cascading is optimal.

- **Users receive sub-optimal utility when the model quality and provider costs rank differently.** Misalignment arises when the models deliver different net value, when churn penalties are low, or when cost-per-success rankings conflict with user utility rankings.

- **Low churn penalties can incentivize providers to throttle latency.** If the likelihood of users in unsubscribing is low, then providers can lower their costs by discouraging repeated use and increasing latency, thereby deteriorating user utility.

## 2    RELATED LITERATURE

**Model routing and cascading.** LLM routing and cascades aim to allocate compute across heterogeneous models to optimize accuracy, latency, and cost for each query (Chen et al., 2023; Ding et al., 2024; Yue et al., 2023; Hu et al., 2024; Dekoninck et al., 2025). Ding et al. (2024) reduce calls to

expensive models by predicting the difficulty of tasks and allocating easier tasks to cheaper models. Cascades extend the routing approach to escalate tasks to stronger models only when cheaper models disagree (Chen et al., 2023; Yue et al., 2023). Dekoninck et al. (2025) propose a unified optimization algorithm for routing and cascading to show optimal routing strategies when the overall quality of a model output with respect to a task can be estimated. This problem has grown recently to include benchmarks for tracking the performance of routers under cost and task completion (Hu et al., 2024). Our paper builds on the behavioral aspect of routing to study when cost-optimal routing acts in a user's best interest. Furthermore, we model the problem by treating the AI interaction via a multi-round prompting game (Castro et al., 2023; Mahmood, 2024), which naturally incorporates the single-round mechanism as a special case.

**Operations and costs of LLM systems.** The deployment of LLM systems must factor for user engagement with respect to performance, costs, and latencies (Bergemann et al., 2025; Zhang et al., 2025). For example, prompt-based interfaces permit users to interact with models via multiple rounds until a task is completed (Bai et al., 2024). Consequently, the cost of a model, is evaluated via a *cost-of-pass*, which tracks the inference cost of a single pass of a model over the probability of the model completing the task in that pass (Mahmood, 2024; Erol et al., 2025). The number of passes in which a user interacts is also highly dependent on model quality with respect to different metrics (Castro et al., 2023). Relatedly, Shirali (2025) study interactive alignment as a Stackelberg game and show how explicit user-side costly signals can reduce the user burden of being understood. Finally, our work ties into the broader literature on game theory for LLM use Sun et al. (2025).

## 3 MAIN PROBLEM

In this section, we formulate a Markov model of the Stackelberg game between an LLM provider routing tasks and a user deciding whether to continue using the LLM. We derive closed-form expressions for expected latencies, utilities, and costs. We motivate with the following example.

**Motivating scenario.** Consider a user of an LLM subscription who is using the LLM to help prove a theorem. The user submits an initial prompt describing the theorem statement and the proof approach they have in mind, and the provider routes the request to either a standard model or a slower reasoning model. After seeing the output, the user either accepts it and stops, or continues the session with a follow-up prompt (e.g., "justify the inequality in line 3") if the proof is incomplete or incorrect. The provider routes the follow-up prompt again, potentially escalating to the reasoning model. If this repeats for multiple prompts, the user may decide to stop outsourcing this task to the LLM; if this experience repeats for multiple theorems, they may eventually cancel the subscription.

### 3.1 MARKOV MODEL OF LLM ROUTING WITH USER RESPONSE

Consider a provider with two LLMs (e.g., standard and reasoning models) denoted $M_i$ for $i \in \{1, 2\}$. A single use of an LLM incurs a monetary cost $c_i \in [0, 1]$ (i.e., inference compute) to the provider and a latency cost $t_i \in [0, 1]$ (i.e., opportunity cost of delay) to the user; we assume the costs are normalized to a standard monetary unit. Given a task from the user, each LLM has a success probability $p_i \in (0, 1)$ of successful completion. We define success as a Bernoulli trial (Mahmood, 2024; Erol et al., 2025). Finally, we assume $t_1 < t_2$, $c_1 < c_2$, and $0 < p_1 < p_2 < 1$.

If an LLM successfully completes a task, the user receives value $V > 0$. Alternatively, if the user abandons the task, the provider incurs a penalty $P > 0$ (e.g., if a user abandons the task, they may unsubscribe from the LLM service entirely with some probability, incurring an expected loss in future revenue for every abandoned task). Finally, we define the user net value per pass $\xi_i := V p_i - t_i$ for each LLM as the expected value received from a single pass of the LLM. We describe $M_i$ as *value-dominated* if $\xi_i > 0$ and *latency-dominated* otherwise.

Figure 2 summarizes the Markov model. At time $\tau$, let $X_\tau \in \{M_1, M_2, \text{Success}, \text{Abandon}\}$ denote the state. Furthermore, let $\tau^*$ be the time of entering one of the two absorption states. Upon receiving a task at $\tau = 0$, the LLM provider commits to a routing and cascading policy $(i, s)$, where $i \in \{1, 2\}$ indicates the initial model to which the task is routed, and $s \in [0, 1]$ is the probability of cascading the task from $M_1$ to $M_2$ if $M_1$ fails to complete the task. Given the provider's policy, the user determines an abandonment policy $q \in [0, 1]$, which is the probability that the user leaves the game without completing the task. We repeat the following sequence:

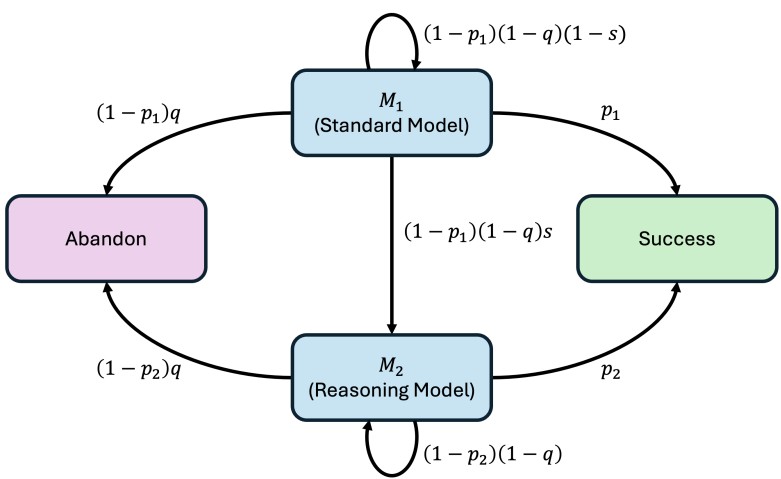

Figure 2: Markov model of LLM provider-user interaction.

1. The provider routes to state $M_i$, incurring user delay $t_i$ and provider cost $c_i$. The model successfully completes the task with probability $p_i$, which transitions to the Success state and yields the user value $V$.

2. If the model fails, the user abandons the task with probability $q$, which transitions to the Abandon state and incurs provider penalty $P$. If the user remains and the provider initially tried $M_1$, they can cascade to $M_2$ with probability $s$ or stay at $M_1$ with probability $1 - s$.

Our key assumption is that the user observes the routing and cascading strategy before deciding a stationary abandonment policy. In practice, users may abandon adaptively with less information. Furthermore, we assume that the success probabilities $p_i$ are i.i.d. for any given task, whereas in practice, the model output and user feedback may allow $p_i$ to change over time. Our assumption follows standard iterative user interaction (Mahmood et al., 2022; Erol et al., 2025). Finally, we note that the provider does not need to observe ground-truth correctness of an output to cascade. Model failure is interpreted through user-side signals (e.g., the user continues the session with follow-up prompts or provides explicit negative feedback), which the provider can act upon (Shirali, 2025).

## 3.2 USER PROBLEM

Let $\alpha(q) := \Pr\{X_{\tau+1} \in \{M_1, M_2\} \mid X_\tau = M_1\} = (1-p_1)(1-q)$ be the probability that $M_1$ fails a task and the user stays. Let $\beta(q) := \mathbb{E}[\sum_{\tau < \tau^*} \mathbb{1}\{X_\tau = M_2\} \mid X_0 = M_2] = 1/(p_2 + (1-p_2)q)$ be the expected number of times that $M_2$ is called before absorption, when initially routed to $M_2$. Furthermore, we define $S_i(s,q) := \Pr\{X_{\tau^*} = \text{Success} \mid i\}$ and $L_i(s,q) := \mathbb{E}[\sum_{\tau < \tau^*} t_\tau \mid i]$ as the success probability and total delay, respectively, for the user given a routing policy $(i, s)$ and abandon policy $q$. That is, $S_i(s,q)$ is the likelihood that the total session eventually absorbs into *Success*, and $L_i(s,q)$ is the expected cumulative latency paid along the sequence of retries and possible escalation. We omit dependency on $s$ and $q$ from these terms when it is obvious. Finally, depending on the initial model, the user success probability and total delay yield closed-form expressions:

$$S_1(s,q) = \frac{p_1 + \alpha(q)\beta(q)p_2 s}{1 - \alpha(q)(1-s)}, \quad S_2(q) = \beta(q)p_2, \quad L_1(s,q) = \frac{t_1 + \alpha(q)\beta(q)t_2 s}{1 - \alpha(q)(1-s)}, \quad L_2(q) = \beta(q)t_2$$

Given provider and user policies, the user's expected utility is the expected value minus total delay

$$U_i(s,q) := \mathbb{E}\left[V\mathbb{1}\{X_\tau = \text{Success}\} - \sum_{\tau < \tau^*} t_\tau \;\middle|\; i\right] = VS_i(s,q) - L_i(s,q)$$

Implicitly, $V$ quantifies the trade-off in utility from successful completion of the task versus the delay from the use of the model. In practice for LLM subscriptions, users pay a fixed fee for use and there is no explicit monetary cost to users per query. As a result, latency is the only marginal cost to the user at interaction time. The user's problem is, given a provider policy $(i, s)$, to determine an abandonment policy that maximizes their utility $q^*(i,s) = \arg\max_q U_i(s,q)$.

### 3.3 PROVIDER PROBLEM

Let $C_i(s,q) := \mathbb{E}[\sum_{\tau<\tau^*} c_\tau \mid i]$ be the service cost incurred to the provider from trying to complete the task, which yields the closed-form expression

$$C_1(s,q) = \frac{c_1 + \alpha(q)\beta(q)c_2 s}{1 - \alpha(q)(1-s)} \qquad C_2(q) = \beta(q)c_2$$

The provider incurs cost equal to the expected service cost plus the penalty from user abandonment

$$J_i(s,q) := \mathbb{E}\left[\sum_{\tau<\tau^*} c_\tau + P\mathbb{1}\{X_\tau = \text{Abandon}\} \;\middle|\; i\right] = C_i(s,q) + P(1 - S_i(s,q))$$

The provider's problem is to route to minimize their cost, assuming the user is maximizing utility, i.e., $(i^*, s^*) = \arg\min_{(i,s)}\{J_i(s, q^*(i,s)) \mid q^*(i,s) = \arg\max_q U_i(s,q)\}$.

## 4 CHARACTERIZING PROVIDER AND USER POLICIES

We characterize the Stackelberg equilibrium by deriving the user best response and the optimal routing policy. When both models are either value- or latency-dominated, user behavior is static and the provider prefers a single, no-escalation route. If the models differ in state, user behavior exhibits threshold structures and the provider's policy almost always reduces to routing without cascading.

### 4.1 USER BEST RESPONSE

We characterize the user best response given a provider policy $(i,s)$. First, when routing to $M_2$, the provider cannot cascade further, and user response depends on if $M_2$ yields positive net value. This can be also interpreted as the user dynamics when faced with a single model in isolation.

**Theorem 1.** *If the provider sets $i = 2$, then the user best response is $q^*(2,s) = \mathbb{1}\{\xi_2 < 0\}$.*

Theorem 1 shows that if the provider immediately routes a task to the better (i.e., reasoning) model, user behavior collapses to determining whether the model is value-dominated (i.e., $\xi_2 > 0$) or latency-dominated (i.e., $\xi_2 < 0$). If the user observes that the delay incurred from using the model is greater than the expected value that the user receives from the model's output to the task, then the user will abandon the task. For example in a coding or mathematics task, the user may find that they can manually solve the problem rather than query a model in the same time. This behavior is characterized by $V$, which is an internal variable qualitatively defined by the user.

We next characterize the user response policy if the provider routes to $M_1$ and provides a cascading policy $s$. Here, the user response depends on the interplay between $\xi_1$ and $\xi_2$.

**Theorem 2.** *If the provider sets $i = 1$, then the user best response depends on $\xi_1$ and $\xi_2$:*

1. *If $\xi_1, \xi_2 \geq 0$, then the user best response is $q^*(1,s) = 0$ for all $s$.*

2. *If $\xi_1, \xi_2 \leq 0$, then the user best response is $q^*(1,s) = 1$ for all $s$.*

3. *If $\xi_1 < 0 < \xi_2$, then let $s_0 := -\xi_1/(\xi_2/p_2 - \xi_1)$. The user best response is $q^*(1,s) = \mathbb{1}\{s < s_0\}$.*

4. *If $\xi_1 > 0 > \xi_2$, then let*

   $$F(s,q) := \xi_1(1-s)\left(p_2 + (1-p_2)q\right)^2 + \xi_2 s\left(1 - (1-p_1)(1-p_2)(1-s)(1-q)^2\right).$$

   *Let $s_L$ be the unique root in $[0,1]$ satisfying $F(s_L, 0) = 0$ and let $s_H := \xi_1/(\xi_1 - \xi_2)$. Then the user best response from routing to model 1 is*

   $$q^*(1,s) = \begin{cases} 0 & \text{if } s \leq s_L \\ q^\dagger(s) & \text{if } s_L < s < s_H \\ 1 & \text{if } s \geq s_H \end{cases}$$

   *where $q^\dagger(s)$ is the unique root in $[0,1]$ to the quadratic equation $F(s, q^\dagger(s)) = 0$.*

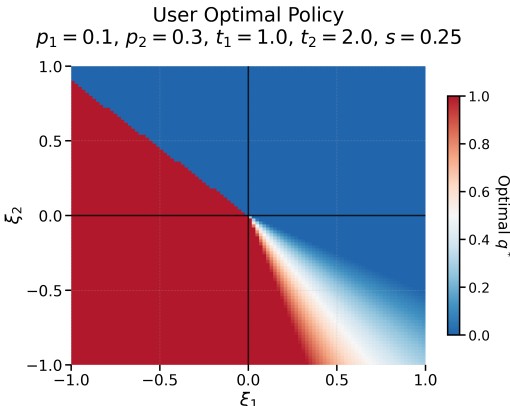

Figure 3: Heatmap of the user response when $i = 1$ and $s = 0.25$. When both models are value-dominated (top right) or latency-dominated (bottom left), the user best response is static. When the models differ in their regime (top left and bottom right), the user response depends on $\xi_1, \xi_2$. Note that $q^* \in (0, 1)$ is in the interior only in certain regimes for $\xi_1 > 0 > \xi_2$.

Figure 3 visualizes the different regimes. From Theorem 2 and Theorem 1, when the two models are undifferentiated in terms of being value- or latency-dominated, then the user response is static. If both models are value-dominated, the user should stay in the system regardless of the routing and cascade policy. If both models are latency-dominated, the user should abandon the task.

Provider routing and cascading policies affect user behavior only when the models offered by the provider are differentiated in value. When $\xi_1 < 0 < \xi_2$, $M_1$ provides negative net value and users should only accept routing to $M_1$ if the likelihood of cascading is high, i.e., $s > s_0$. Otherwise, users can abandon the task since the expected delay is too large, replicating the single-model logic of Theorem 1. Finally, if $\xi_1 > 0 > \xi_2$, $M_1$ is value-dominated and $M_2$ is latency-dominated. Here, routing directly to $M_2$ should encourage users to abandon the task. If the provider first routes to $M_1$ and assigns a low $s < s_L$, then the user is incentivized to remain. Raising $s > s_H$ pushes users to abandon the task since it may get cascaded to a latency-dominated $M_2$. Most interestingly, the interior regime $(s_L, s_H)$ has a non-static $q^*(1, s) = q^\dagger(s)$, where the user is incentivized to abandon the task with probability. This is the only scenario where user behavior is stochastic and changes with slight perturbation of $s$. Although the provider-optimal policy may find $s \in (s_L, s_H)$, it may be also strategic to avoid this regime for a provider to ensure user behavior is predictable.

Theorem 2 assumes that users know $(i, s, p_1, p_2)$. In practice, the LLM interface may reveal the route $i$, but not $s$, $p_1$, or $p_2$. Here, beliefs drive the same threshold logic with respect to belief summaries; for example, users can act using $\mathbb{E}_\pi[s]$ where $\pi(s)$ is a posterior estimate of $s$ observed from past routing. Over repeated tasks, this leads to an exploration versus exploitation problem.

## 4.2 PROVIDER-OPTIMAL POLICY

Given the user best response, the provider optimization problem becomes taking the maximum of a single-variable optimization problem and a scalar:

$$\min \left\{ \min_s J_1\left(s, q^*(1, s)\right),\ J_2\left(\mathbb{1}\{\xi_2 < 0\}\right) \right\}$$

The structure of $J_1(s, q^*(1, s))$ depends on which models are value- or latency-dominated. We first prove the two cases where the models share the same sign with respect to $\xi_i$, before considering the settings where the models are differentiated.

**Theorem 3.** *When both models share the same sign, the provider-optimal policy involves routing to one model without cascading:*

1. *If $\xi_1, \xi_2 > 0$, then*

$$(i^*, s^*) = \begin{cases} (1, 0) & \text{if } \dfrac{c_1}{p_1} \le \dfrac{c_2}{p_2} \\ (2, 0) & \text{otherwise} \end{cases}$$

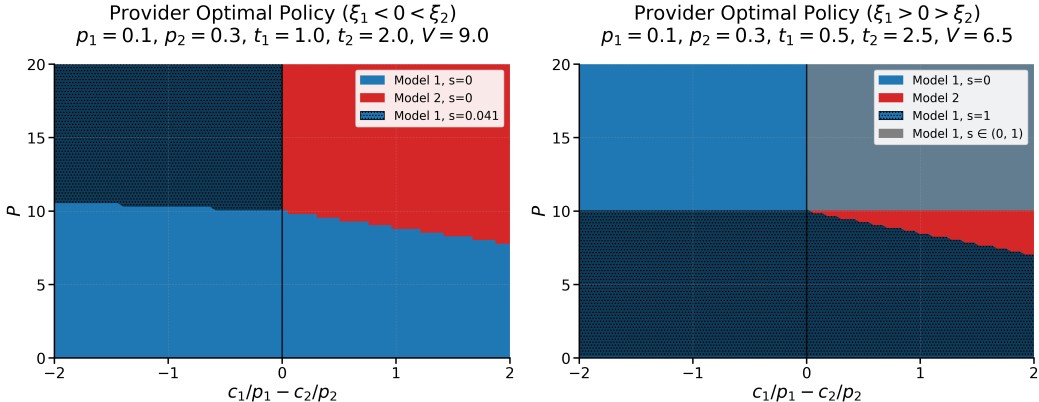

Figure 4: *Left:* Heatmap of the provider-optimal policy for $\xi_1 < 0 < \xi_2$. *Right:* Heatmap of the provider-optimal policy for $\xi_1 > 0 > \xi_2$. For both plots, we hold $c_1 = 1$ constant and sweep the difference in cost-of-pass $c_1/p_1 - c_2/p_2$ as well as $P$. For most regimes, the optimal policy is either to route immediately to $M_1$ or to $M_2$ without any cascading. There exist only some regimes for $\xi_1 > 0 > \xi_2$ where the optimal policy involves probabilistic $s \in (0, 1)$.

2. If $\xi_1, \xi_2 < 0$, then, setting $s \in [0, 1]$ as a free variable,

$$(i^*, s^*) = \begin{cases} (1, s) & \text{if } P \le \dfrac{c_2 - c_1}{p_2 - p_1} \\ (2, 0) & \text{otherwise} \end{cases}$$

If the two models are undifferentiated, i.e., both value- or latency-dominated, then cascading from one model to the other adds costs and variance without any benefits, making single routing optimal. When both models are value-dominated, the optimal policy is to route to the best model determined by the cost-of-pass (Mahmood, 2024; Erol et al., 2025). Because users should not abandon tasks, the penalty for user abandonment is irrelevant and cascading cannot improve costs beyond routing to the appropriate model immediately. When both models are latency-dominated, users are likely to abandon tasks if the model cannot complete them. Here, the routing policy depends on $P$ versus the incremental advantage in the cost-of-pass $(c_2 - c_1)/(p_2 - p_1)$. If the penalty of user abandonment is low, the optimal provider policy is to route to the cheaper standard model. Moreover, the choice of $s$ is irrelevant since users will abandon regardless. On the other hand, if $P$ is large, then the provider must route to the reasoning model to reduce the risk of incurring the penalty.

**Theorem 4.** *Let $P_1 := (c_2/p_2 - c_1)/(1 - p_1)$ and let $P_2 := (c_1(1 - s_0) + c_2 s_0/p_2)/(p_1 + (1 - p_1)s_0)$. If $\xi_1 < 0 < \xi_2$, then the provider-optimal policy breaks into four cases:*

$$(i^*, s^*) = \begin{cases} (1, 0) & \text{if } \dfrac{c_1}{p_1} > \dfrac{c_2}{p_2} \text{ and } P < P_1 \\ (2, 0) & \text{if } \dfrac{c_1}{p_1} > \dfrac{c_2}{p_2} \text{ and } P > P_1 \\ (1, 0) & \text{if } \dfrac{c_1}{p_1} < \dfrac{c_2}{p_2} \text{ and } P < P_2 \\ (1, s_0) & \text{if } \dfrac{c_1}{p_1} < \dfrac{c_2}{p_2} \text{ and } P > P_2 \end{cases}$$

When $\xi_1 < 0 < \xi_2$, users should remain if routed to $M_2$, but leave if routed to $M_1$ with limited chance of escalating to $M_2$. However, if $c_1/p_1 < c_2/p_2$, then the provider should always route to $M_1$. Alternatively if $P < P_1$ for the threshold value $P_1$, then the provider should still route to $M_1$. Interestingly, even though the reasoning model is preferable to users, the provider-optimal policy is to route to $M_1$, unless the cost-of-pass of this model is sufficiently high. This leads to misalignment between the user and the provider when their respective rankings of the models disagree.

Finally, we consider the case when $\xi_1 > 0 > \xi_2$. Here, the user best response has three regions defined by thresholds $s_L$ and $s_H$. Given the monotonicity of the provider function, determining the optimal routing and cascade policy is equivalent to optimizing over the following five settings:

$$\min \left\{ J_1(0, 0), J_1(s_L, 0), \min_{s \in (s_L, s_H)} J_1(s, q^\dagger(s)), J_1(1, 1), J_2(0) \right\}$$

Because the provider policy is not necessarily static in this regime, we focus on sufficient conditions under which the provider can employ static policies.

**Theorem 5.** *If $\xi_1 > 0 > \xi_2$, the following statements are true:*

1. *If $c_1/p_1 < \min\{P, c_2/p_2\}$, the provider-optimal policy is to route to model 1 only $(i^*, s^*) = (1, 0)$.*

2. *If $P < \min\{c_1/p_1, (c_2 - c_1)/(p_2 - p_1)\}$, the provider-optimal policy is to route to model 1 and cascade on failure $(i^*, s^*) = (1, s)$ where $s \in [s_H, 1]$ is a free variable.*

3. *If $(c_2 - c_1)/(p_2 - p_1) < P < c_2/p_2 < c_1/p_1$, the provider-optimal policy is to route to model 2 $(i^*, s^*) = (2, 0)$.*

Figure 4 visualizes the two settings where $\xi_1 < 0 < \xi_2$ and $\xi_1 > 0 > \xi_2$, respectively. For most settings, a static policy, i.e., $(i^*, s^*) \in \{(1, 0), (1, 1), (2, 0)\}$, is optimal. Moreover in both cases, the optimal policy is almost always to route to $M_1$ first. This underscores the value of routing to a weaker model initially, when the models are differentiated. Furthermore, the role of cascading differs depending on which of the two models is value-dominated. If $M_1$ is the only value-dominated model, then cascading is effective specifically when the penalty is sufficiently low with respect to $c_1/p_1$; otherwise, the provider is incentivized to remain at the standard model. If $M_2$ is value-dominated, then aggressive cascading via $s = 1$ is useful only when $P$ is sufficiently small.

Finally, we note that although we fix $V$, $p_1$, and $p_2$, the results naturally extend to a heterogenous setting with task-dependent parameters. Given a prompt $x$, a routing algorithm may first estimate the success probability $p_i(x)$ and user value $V(x)$ to better estimate the per-task user value $\xi_i(x) = V(x)p_i(x) - t_i$ and determine a personalized optimal route.

## 5 WHEN ARE PROVIDER AND USER MISALIGNED?

In this section, we discuss the implications of the equilibrium between user and provider to understand when the provider routing decisions act in the best interest of the user. We first define a misalignment gap that characterizes when the provider's cost-minimizing routing strategy is suboptimal for the user utility function. We then study the conditions when misalignment exists.

Consider the cost-optimal policy for a provider $(i^*, s^*)$. The misalignment gap is the difference between a user utility-optimal routing policy versus the provider's true policy

$$\Delta_U := \max_{i, s} U_i(s, q^*(i, s)) - U_{i^*}(s^*, q^*(i^*, s^*))$$

If $\Delta_U = 0$, then the provider-optimal policy is also optimal for the user, meaning that the provider's actions are in the user's best interests. However, if $\Delta_U > 0$, then there is a utility gap for the user incurred by the provider minimizing service costs. Depending on the costs and penalties, a misalignment gap can exist for every possible configuration of $\xi_1, \xi_2$.

**Proposition 1.** *The following statements are true:*

1. *If $\xi_1, \xi_2 > 0$, then $\Delta_U = 0$ if and only if $\text{sign}(c_1/p_1 - c_2/p_2) = \text{sign}(\xi_2/p_2 - \xi_1/p_1)$.*

2. *If $\xi_1, \xi_2 < 0$, then $\Delta_U = 0$ if and only if $\text{sign}(\xi_2 - \xi_1) = \text{sign}(P - (c_2 - c_1)/(p_2 - p_1))$.*

3. *If $\text{sign}(\xi_1) \neq \text{sign}(\xi_2)$, then $\Delta_U = 0$ if and only if $\xi_{i^*} > 0$ and $s^* = 0$.*

When $\xi_1, \xi_2 > 0$, provider routing collapses to a static cost-of-pass rule. Here, alignment requires the provider and user rankings to match; otherwise the provider selects the cheaper model while the user prefers the more successful one. When $\xi_1, \xi_2 < 0$, the user always abandons, meaning that alignment requires the provider to select the same "less-bad" model from the user's perspective. This only occurs if the provider's switch regime agrees with the user's ranking. Finally if the models are differentiated, any likelihood of cascading exposes the user to latency. Figure 5 (Left, Middle) highlight this regime for the same parameters as in Figure 4, showing the misalignment gap.

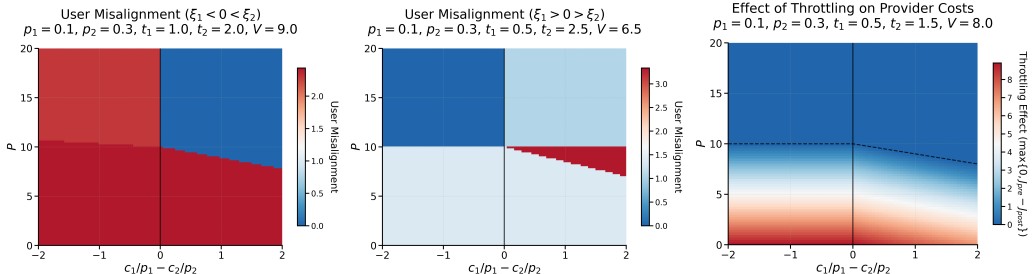

Figure 5: *Left:* Heatmap of the user misalignment gap for $\xi_1 < 0 < \xi_2$. *Middle:* Heatmap of the user misalignment gap for $\xi_1 > 0 > \xi_2$. *Right:* Heatmap of the effect of throttling on provider costs; the dashed line is the line $P = \min\{c_1/p_1, c_2/p_2\}$. For all plots, we hold $c_1 = 1$ constant and sweep the difference in cost-of-pass $c_1/p_1 - c_2/p_2$ as well as $P$.

## 6    THE RISK OF THROTTLING LATENCY

Finally, we study an extreme case of misalignment where the provider may seek to inflate the latency of their models to encourage users to abandon tasks more often, thereby reducing service costs from repeated prompts. Specifically, consider a provider forced to a fixed subscription price due to factors such as market competition, meaning they cannot directly re-price in response to higher service costs. Instead, throttling latency functions as a hidden price increase that implicitly reduces usage, and consequently, provider costs. Here, user abandonment causes a penalty $P$, but if it is sufficiently small, this loss may offset the higher service costs.

Suppose the models have latencies $t_i < Vp_i$ for $i \in \{1, 2\}$. Let $q^*(i, s; t_1, t_2)$ be the user response for these models and let $J^*_{pre} := \min_{(i,s)}\{J_i(s, q^*(i, s; t_1, t_2))\}$ be the provider-optimal cost. Given two *throttled* latencies $\hat{t}_i > t_i$ for $i \in \{1, 2\}$, let $J^*_{post} := \min_{(i,s)}\{J_i(s, q^*(i, s; \hat{t}_1, \hat{t}_2))\}$ as the provider-optimal cost after throttling. Note that $\hat{t}_i$ are now variables that the provider can tune. If $J^*_{post} \leq J^*_{pre}$, then the provider benefits from throttling, even though it reduces the user's utility.

**Proposition 2.** *If $P \leq \min\{c_1/p_1, c_2/p_2\}$, then setting $\hat{t}_i > Vp_i$ incurs $J^*_{post} \leq J^*_{pre}$.*

Figure 5 (Right) visualizes the gain from throttling. For static policies, the gain is $\min_i\{c_i/p_i\} - \min_i\{c_i + P(1 - p_i)\}$, which is linear in $P$. Moreover, throttling maximizes misalignment $\Delta_U$, since providers prefer users to abandon tasks while users would prefer to successfully complete tasks. Most importantly, providers will reduce costs from throttling either (or both) models.

When $P$ exceeds $\min\{c_1/p_1, c_2/p_2\}$, the gain from throttling becomes negative and this misaligned policy loses its dominance. Therefore, to prevent throttling, the price of user abandonment must exceed the cost-per-pass. In practice, this can be realized either by users proactively unsubscribing from LLM providers, by providers voluntarily ensuring a guarantee to users, or by permitting users to opt-out of routing and cascading by self-selecting models.

## 7    CONCLUSION

We study the problem of LLM routing as an interaction between a provider offering a menu of models and a user subscribing to the LLM service seeking the best model to complete their task. The user impact of routing depends on the patience of users facing the trade-offs between quality, latency, and cost. We formalize a Stackelberg game between a single provider with two models, standard and reasoning, and a single user. The user maximizes their utility from LLM use by abandoning tasks with large inference delay, while the provider minimizes the cost of serving the user and the risk of users leaving the LLM subscription service when they abandon.

We find that user patience for the LLM provider is governed by the sign of net value the user receives from each model. If both models offer similar value, then the user's action is unaffected by the routing policy. On the other hand, if the two models differ in value, then users are incentivized to abandon their tasks with some probability if the likelihood of being routed or cascaded to the worse model is high. On the other hand, the optimal provider routing policy is almost always static, showing that there is limited value in cascading except in a narrow regime where the models are

differentiated in user net value. This leads to a provider-user misalignment when the value rankings from a user and the cost-of-pass rankings from a provider disagree. Most importantly, misalignment can grow extreme when users are unlikely to unsubscribe. This motivates providers to artificially throttle latency, which saves their costs but yields poorer user value.

The extant routing literature often optimizes over the trade-off between solution quality and latency by estimated per-task success rates (Hu et al., 2024). Our analysis suggests that user behavior can play a core role in determining optimal routing policies. For instance, the development of this paper involved sustained use of an LLM service, where for example, deriving key theoretic results (i.e., high-value tasks) necessitated user patience from repeated prompts (see Appendix A for details). Consequently, we advocate for LLM routing algorithms to simultaneously estimate both model success rates and user value per task in order to better serve users.

We note several limitations and next steps. First, our analysis is of two models, but the framework can be extended to more models. Second, our subscription framing abstracts away per-query monetary prices; pay-per-call API pricing can be incorporated by adding a per-query charge to the user utility. Third, we assume that a user can observe the provider cascade strategy and acts with a stationary abandonment policy. In practice, routing strategies are hidden and users may need to adapt. Finally, we treat the per-pass success as i.i.d. and assume fixed user value and provider penalty that are known to both players. In practice, these terms may not always be immediately quantifiable and may require online learning.

ACKNOWLEDGMENTS

The author thanks the ICLR 2026 editorial chairs and anonymous referees for providing valuable feedback that improved the content of this paper. This work was funded partially by the Natural Sciences and Engineering Research Council of Canada.

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

| Task type | LLM role | Acceptance criterion | Typical outcome |
|---|---|---|---|
| Proof sketching | Suggest directions, arguments, and counterexamples | Human derivation reproduces all steps | Often useful after 2–10 reprompts |
| Algebraic checking | Expand derivatives, monotonicity, or bounding steps | Symbols match manual calculations | Often useful after 2–4 reprompts |
| Draft writing | Prepare bullet form for text, condense paragraphs and enforce writing style | Preserves technical meaning and improves clarity | Useful with one pass |
| Figure prototyping | Generate plotting scaffolds | Matches definitions and theorem regimes | Useful with minor edits |

Table 1: Summary of how LLM assistance was used and validated.

## A  PROTOCOL OF LLM USAGE

We used LLMs as assistive tools throughout this project. The assistance covered: (i) deriving proof steps or searching for counterexamples; (ii) preparing first drafts for sections and editing with feedback; and (iii) coding plots. All mathematical statements, derivations, and proofs in the final paper were human-verified end-to-end. All figures and text underwent human editing.

We interacted with the LLM in multi-round sessions. Each session targeted a single goal (e.g., proof sketch, draft writing and editing, or code). For many theoretical results, although we were routed to the complex reasoning model, we found our user behavior to be patient in repeated interactions, suggesting the high user value placed on the work. For certain tasks and over time, we found our behavior to be less patient, suggesting a time-varying nature on user value, which could be studied in future work. Table 1 summarizes typical tasks, the intended role of the LLM, and acceptance criteria.

# B    HELPER LEMMAS

In this section, we outline several lemmas required for the main proofs of the paper.

**Lemma 1.** *For any fixed $q$, the objective function for the provider is monotone in $s$.*

*Proof of Lemma 1.* It suffices to prove that $C_1(s)$ and $S_1(s)$ are monotone in $s$. Note that both $C_1(s)$ and $S_1(s)$ can be written as fractions of two affine functions in $s$. As a result, the first derivative of these functions is a ratio of a constant value over a non-negative quadratic function $(1 - \alpha(1 - s))^2$. $\qquad\square$

**Lemma 2.** *For any fixed $s$, let $U_1^+(\alpha) := (\xi_1 + \xi_2\alpha s/p_2)/(1 - \alpha(1 - s))$ and let $U_1^-(\alpha) := (\xi_1 + \xi_2\alpha s)/(1 - \alpha(1 - s))$. Then for all $q \in [0,1]$,*

$$U_1^-(\alpha) \le U_1(s,q) \le U_1^+(\alpha) \qquad \textit{if } \xi_2 \ge 0$$
$$U_1^+(\alpha) \le U_1(s,q) \le U_1^-(\alpha) \qquad \textit{otherwise}$$

*Proof of Lemma 2.* Note that $\beta(q)$ is a decreasing function of $q$ and is bounded $[1, 1/p_2]$. Therefore, for any fixed $s, q$, if $\xi_2 \ge 0$, then $\xi_1 + \xi_2\alpha s \le \xi_1 + \xi_2\alpha\beta s \le \xi_1 + \xi_2\alpha s/p_2$. Moreover, the inequalities are reversed when $\xi_2 < 0$. $\qquad\square$

**Lemma 3.** *Let $K^+(s) := \xi_1(1 - s) + \xi_2 s/p_2$ and let $K^-(s) := \xi_1(1 - s) + \xi_2 s$.*

1. *$U_1^+(\alpha)$ is monotone increasing (respectively, decreasing) in $\alpha$ and therefore, decreasing (respectively, increasing) in $q$, if $K^+(s) > 0$ (respectively, if $K^+(s) < 0$).*

2. *$U_1^-(\alpha)$ is monotone increasing (respectively, decreasing) in $\alpha$ and therefore, decreasing (respectively, increasing) in $q$, if $K^-(s) > 0$ (respectively, if $K^-(s) < 0$).*

*Proof of Lemma 3.*

1. We take the first derivative of $U_1^+$ as follows

$$\frac{dU_1^+}{d\alpha} = \frac{\xi_2(1 - \alpha(1 - s))/p_2 + (1 - s)(\xi_1 + \xi_2\alpha s/p_2)}{(1 - \alpha(1 - s))^2} = \frac{K^+(s)}{(1 - \alpha(1 - s))^2}$$

   Since the denominator is positive for all $s \in [0,1]$, the first derivative is strictly positive if $K^+(s) > 0$ and strictly negative otherwise. Consequently, $U_1^+(s)$ is increasing or decreasing, respectively, in $\alpha$. Finally, note that $\alpha(q)$ is a decreasing function in $q$, meaning that $U_1^+(\alpha(q))$ is decreasing or increasing, respectively in $q$.

2. We take the first derivative $dU_1^-/d\alpha = K^-(s)/(1 - \alpha(1 - s))^2$ and use the same steps to show $U_1^-(\alpha)$ is increasing in $\alpha$ and therefore, decreasing in $q$ if $K^-(s) > 0$ and the reverse otherwise.

$\qquad\square$

**Lemma 4.** *Let $P_1 := (c_2/p_2 - c_1)/(1 - p_1)$. If $c_1/p_1 > c_2/p_2$, then $P_1 > (c_2 - c_1)/(p_2 - p_1)$. Otherwise, $P_1 \le (c_2 - c_1)/(p_2 - p_1)$.*

*Proof.* Proof of Lemma 4 We subtract

$$P_1 - \frac{c_2 - c_1}{p_2 - p_1} = \frac{\left(\frac{c_2}{p_2} - c_1\right)(p_2 - p_1) - (c_2 - c_1)(1 - p_1)}{(1 - p_1)(p_2 - p_1)} = \frac{(1 - p_2)\left(c_1 - \frac{c_2 p_1}{p_2}\right)}{(1 - p_1)(p_2 - p_1)}.$$

From the numerator, this fraction is positive when $c_1/p_1 > c_2/p_2$, and is non-positive otherwise. $\qquad\square$

**Lemma 5.** *Let $P_2 := (c_1(1 - s_0) + c_2 s_0/p_2)/(p_1 + (1 - p_1)s_0)$. If $c_1/p_1 < c_2/p_2$, then $P_2 \in (c_1/p_1, c_2/p_2)$.*

*Proof.* Proof of Lemma 5 We will show below that $P_2$ is a convex combination of $c_1/p_1$ and $c_2/p_2$. Consider the general function

$$P_2(s) := \frac{c_1(1-s) + \frac{c_2}{p_2}s}{p_1 + (1-p_1)s} \quad \implies \quad \frac{dP_2}{ds} = \frac{\frac{c_2 p_1}{p_2} - c_1}{(p_1 + (1-p_1)s)^2}$$

Note that when $c_1/p_1 < c_2/p_2$, this function $P_2(s)$ is monotone increasing. Furthermore, $P_2(0) = c_1/p_1$ and $P_2(1) = c_2/p_2$. Since $s_0 \in (0,1)$, $P_2(s_0) =: P_2$ represents a convex combination. $\square$

**Lemma 6.** *The following identities hold for all $s, q \in [0, 1]$:*

1. $S_1(s, q) \geq p_1$

2. $\min\{c_1/p_1, \; c_2/p_2\} \leq (c_1(1-s) + \beta(q)c_2 s)/(p_1(1-s) + \beta(q)p_2 s) \leq \max\{c_1/p_1, \; c_2/p_2\}$

*Proof of Lemma 6.*

1. We subtract as follows:

$$S_1(s, q) - p_1 = \frac{p_1 + \alpha(q)\beta(q)p_2 s}{1 - \alpha(q)(1-s)} - p_1 = \frac{\alpha(q)\left(\beta(q)p_2 s + p_1(1-s)\right)}{1 - \alpha(q)(1-s)} \geq 0 \qquad \forall s, q \in [0, 1]$$

2. Let $R(s) := (c_1(1-s) + \beta(q)c_2 s)/(p_1(1-s) + \beta(q)p_2 s)$. Note that

$$\frac{dR}{ds} = \frac{\beta(q)\left(c_2 p_1 - p_2 c_1\right)}{p_1 + s(\beta(q)p_2 - p_1)},$$

meaning that $R(s)$ is monotonic increasing when $c_2/p_2 > c_1/p_1$ and is monotonic non-increasing otherwise. Furthermore, note that $R(0) = c_1/p_1$ and $R(1) = c_2/p_2$. Therefore, $R(s)$ is a convex combination of $c_1/p_1$ and $c_2/p_2$.

$\square$

# C PROOFS OF MAIN RESULTS

## C.1 PROOFS FOR SECTION 4.1

*Proof of Theorem 1.* We take the first derivative of $U_2(q)$ to show

$$\frac{dU_2}{dq} = -\frac{\xi_2(1-p_2)}{(p_2 + (1-p_2)q)^2}$$

that the derivative is positive for all $q \in [0,1]$ if $\xi_2 < 0$ and is non-positive otherwise. If the derivative is positive, then $U_2(q)$ is monotone increasing and the optimal response is $q^* = 1$. If the derivative is non-positive, then the utility is non-increasing and an optimal response is $q^* = 0$. □

*Proof of Theorem 2.*

1. **If $\xi_1, \xi_2 \geq 0$:** From Lemma 2, $U_1(s,q) \leq U_1^+(\alpha)$. Furthermore because $\xi_1 \geq 0$, from Lemma 3, $K^+(s) \geq 0$ for all $s \in [0,1]$ and $U_1^+(\alpha)$ is monotone non-decreasing in $q$. Consequently,

$$U_1(s,q) \leq U_1^+(\alpha) \leq U_1^+(\alpha(0)) = \frac{\xi_1 + \xi_2 s \frac{1-p_1}{p_2}}{1 - (1-p_1)(1-s)} = U_1(s,0)$$

   showing that the envelope function is tight at the maximizing point $q^* = 0$.

2. **If $\xi_1, \xi_2 < 0$:** From Lemma 2, $U_1(s,q) \leq U_1^-(\alpha)$. Furthermore, because $\xi_1 \leq 0$, from Lemma 3, $K^-(s) < 0$ for all $s \in [0,1]$ and $U_1^-(\alpha)$ is monotone non-increasing in $q$. Consequently,

$$U_1(s,q) \leq U_1^-(\alpha) \leq U_1^-(\alpha(1)) = \xi_1 = U_1(s,1)$$

   showing that the envelope function is tight at the maximizing point $q^* = 1$.

3. **If $\xi_1 < 0 < \xi_2$:** From Lemma 2, $U_1(s,q) \leq U_1^+(\alpha)$. Furthermore from Lemma 3, the envelope function is monotone decreasing in $q$ if $K^+(s) > 0$ and monotone non-decreasing otherwise. Since $K^+(s)$ is a convex combination of a positive and negative value, we have $K^+(s) > 0$ for all $s > s_0$ and $K^+(s) \leq 0$ otherwise.

   We first consider the case of $s > s_0$. Here, we leverage the monotone decreasing property of the envelope

$$U_1(s,q) \leq U_1^+(\alpha) \leq U_1^+(\alpha(0)) = \frac{\xi_1 + \xi_2 s \frac{1-p_1}{p_2}}{1 - (1-p_1)(1-s)} = U_1(s,1)$$

   Similarly, when $s < s_0$, we leverage the monotone increasing property of the envelope

$$U_1(s,q) \leq U_1^+(\alpha) \leq U_1^+(\alpha(1)) = \xi_1 = U_1(s,1)$$

   Both cases show that the envelope function is tight at their respective maximizing points.

4. **If $\xi_1 > 0 > \xi_2$:** We break the proof into three regimes for $s \geq s_H$, $s \leq s_L$, and $s \in (s_L, s_H)$.

   **For $s \geq s_H$:** We first characterize the user best response using the envelope arguments. From Lemma 2 and Lemma 3, we have $U_1(s,q) \leq U_1^-(\alpha)$, which is monotone non-increasing if and only if $K_{(}^- s) \leq 0$. This occurs only in the regime for $s \geq s_H$. Here,

$$U_1(s,q) \leq U_1^-(\alpha) \leq U_1^-(\alpha(1)) = \xi_1 = U_1(s,1)$$

   showing the envelope function is tight at the maximizing point.

   **For $s \leq s_L$:** For the remaining two regimes, the envelope will not be a tight upper bound, meaning this argument will not be viable and we require an alternate approach.

For notational simplicity, let $U_1(s,q) = N(s,q)/D(s,q)$ where $N(s,q) := \xi_1 + \xi_2 \alpha(q)\beta(q)s$ and $D(s,q) = 1 - \alpha(q)(1-s)$. We seek to identify a constant $s_L$ such that for any $s < s_L$,

$$U_1(s,q) - U_1(s,0) = \frac{N(s,q)D(s,0) - N(s,0)D(s,q)}{D(s,q)D(s,0)} \leq 0 \qquad \forall q \in [0,1]$$

Since the denominator of this fraction is always positive, we consider the numerator

$$N(s,q)D(s,0) - N(s,0)D(s,q) \tag{1}$$

$$= \left( \xi_1 + \frac{(1-p_1)(1-q)}{p_2 + (1-p_2)q} \xi_2 s \right) (p_1 + (1-p_1)s)$$

$$- \left( \xi_1 + \frac{1-p_1}{p_2} \xi_2 s \right) (p_1 + (1-p_1)s + (1-p_1)(1-s)q) \tag{2}$$

$$= \xi_2 s(1-p_1)(p_1 + (1-p_1)s) \left( \frac{1-q}{p_2 + (1-p_2)q} - \frac{1}{p_2} \right) - (1-p_1)(1-s)q \left( \xi_1 + \frac{1-p_1}{p_2} \xi_2 s \right) \tag{3}$$

$$= -\xi_2 sq(1-p_1) \frac{p_1 + (1-p_1)s}{(p_2 + (1-p_2)q)p_2} - (1-p_1)(1-s)q \left( \xi_1 + \frac{1-p_1}{p_2} \xi_2 s \right) \tag{4}$$

$$= -q(1-p_1) \left( \frac{p_1 + (1-p_1)s}{p_2 + (1-p_2)q} \frac{\xi_2 s}{p_2} + \xi_1(1-s) + (1-p_1)(1-s) \frac{\xi_2 s}{p_2} \right) \tag{5}$$

$$= -q(1-p_1) \underbrace{\left( \xi_1(1-s) + \frac{\xi_2 s}{p_2} \left( \frac{p_1 + (1-p_1)s}{p_2 + (1-p_2)q} + (1-p_1)(1-s) \right) \right)}_{=:T(s,q)} \tag{6}$$

Above, (3) cancels $\xi_1(p_1 + (1-p_1)s)$ and groups the terms, (4) combines the fractions in parentheses, and (5) factors $q(1-p_1)$. For notational simplicity, let $T(s,q)$ be the equation within the parentheses of the final equality. Note that $p_2^2 T(s,0) = F(s,0)$ for $F(s,q)$ as defined in the theorem statement. Rather than characterizing $s_L$ as the root of $F(s_L,0) = 0$, we will characterize it as the root of $T(s_L,0) = 0$.

Given the definition of $T(s,q)$, we observe that for $U_1(s,q) - U_1(s,0) \leq 0$ for all $q \in [0,1]$, we must have

$$T(s,q) \geq \min_{q \in [0,1]} T(s,q) \geq 0.$$

In order to find the minimizing $q$, we take the derivative below

$$\frac{\partial T(s,q)}{\partial q} = -\frac{(p_1 + (1-p_1)s)(1-p_2)\xi_2 s}{(p_2 + (1-p_2)q)^2 p_2} \geq 0$$

where the inequality follows from $\xi_2 < 0$. Because the derivative is always non-negative, $T(s,q)$ is monotone non-increasing and the function is minimized at $q = 0$.

We now must find a value of $s_L$ such that for all $s < s_L$, we have $T(s,q) \geq 0$, which implies further that $U_1(s,q) - U_1(s,0) \geq 0$ for all $q \in [0,1]$. Note that $T(0,0) = \xi_1 > 0$ and $T(1,0) = \xi_2/p_2^2 < 0$. Furthermore, the derivative

$$\frac{dT}{ds} = -\xi_1 + \frac{\xi_2}{p_2} \left( \frac{p_1}{p_2} + 1 - p_1 \right) + \frac{2\xi_2(1-p_1)(1-p_2)}{p_2^2} < 0$$

meaning that for the interval $[0,1]$, $T(s,0)$ begins positive, decreases monotonically, and ends negative. Therefore, it must have a unique root within the interval $[0,1]$. Let $s_L$ be this root. Thus, for all $s < s_L$, $T(s,0)$ is positive and the user best response $q^*(1,s) = 0$.

**For $s \in (s_L, s_H)$:** Here, we directly obtain the the optimal $q^*(1,s)$ via the first-order optimality condition $0 = \partial U_1/\partial q = (N'(s,q)D(s,q) - N(s,q)D'(s,q))/D(s,q)^2$. Because the denominator is always positive, this condition is equivalent to

$$0 = N'(s,q)D(s,q) - N(s,q)D'(s,q)$$

$$= -\frac{(1-p_1)\xi_2 s}{(p_2 + (1-p_2)q)^2} (1 - (1-p_1)(1-q)(1-s)) - \left( \xi_1 + \frac{(1-p_1)(1-q)\xi_2 s}{p_2 + (1-p_2)q} \right) (1-p_1)(1-s)$$

By factoring out $-(1 - p_1)/(p_2 + (1 - p_2)q)^2$, we simplify our condition to

$$
\begin{aligned}
0 &= \xi_2 s(1 - (1 - p_1)(1 - q)(1 - s)) + \xi_1(1 - s)(p_2 + (1 - p_2)q)^2 \\
&\quad + \xi_2 s(1 - s)(1 - p_1)(1 - q)(p_2 + (1 - p_2)q) \tag{7} \\
&= \xi_1(1 - s)(p_2 + (1 - p_2)q)^2 + \xi_2 s(1 - (1 - p_1)(1 - p_2)(1 - s)(1 - q)^2) \tag{8} \\
&= \xi_1(1 - s)p_2^2 + \xi_2 s(1 - (1 - p_1)(1 - p_2)(1 - s)) \\
&\quad + 2q(1 - s)(1 - p_2)\left(\xi_1 p_2 + \xi_2 s(1 - p_1)\right) \\
&\quad + q^2(1 - s)(1 - p_2)\left(\xi_1(1 - p_2) - \xi_2 s(1 - p_1)\right) \tag{9}
\end{aligned}
$$

where (7)–(9) follow from expanding and collecting like terms. Let $F(s, q) :=$ RHS (9) be the implicit function defining this equation. We must now prove that for any $s \in (s_L, s_H)$, there exists a unique $q^\dagger(s)$ that satisfies $F(s, q^\dagger(s)) = 0$.

To prove that a unique root always exists, we first observe that for any fixed $s$, $F(s, q)$ is a convex function in $q$, since the slope of the first partial derivative is $2(1-s)(1-p_2)(\xi_1(1-p_2) - \xi_2 s(1-p_1)) \geq 0$. Furthermore, we show that that $F(s, q)$ takes different signs at the boundaries, notably $\lim_{s \to s_L^+, q \to 0} F(s, q) < 0$ and $\lim_{s \to s_H^-, q \to 1} F(s, q) > 0$, where at the limits, the user response moves towards the optimal boundary solutions.

For the lower limit

$$
\begin{aligned}
\lim_{s \to s_L^+, q \to 0} F(s, q) &\propto - \lim_{s \to s_L^+, q \to 0} \frac{\partial U_1}{\partial q} \\
&= - \lim_{s \to s_L^+, q \to 0} \frac{U_1(s, q) - U_1(s, 0)}{q} \\
&= - \lim_{s \to s_L^+, q \to 0} \frac{-(1 - p_1)T(s, q)q}{D(s, 0)D(s, q)q} \\
&= \lim_{s \to s_L^+, q \to 0} \frac{(1 - p_1)T(s, q)}{D(s, q)^2} \\
&< 0 \ \text{ for } s > s_L.
\end{aligned}
$$

where the inequality arises from the observation that $T(s) > 0$ in this regime.

For the upper limit

$$
\begin{aligned}
\lim_{s \to s_H^- q \to 1} F(s, q) &\propto - \lim_{s \to s_H^-, q \to 1} \frac{\partial U_1}{\partial q} \\
&= - \lim_{s \to s_H^-, q \to 1} \frac{N'(s, q)D(s, q) - N(s, q)D'(s, q)}{D(s, q)^2} \\
&= \lim_{s \to s_H^-} (1 - p_1)\left(\xi_1(1 - s) + \xi_2 s\right) \\
&=> 0 \ \text{ for } s < s_H.
\end{aligned}
$$

By the Intermediate Value Theorem, for any fixed $s \in (s_L, s_H)$, the function $F(s, q) = 0$ passes for some $q$. Because $F(s, q)$ is a quadratic function of $q$, it can change direction at most once and therefore, the root $q^\dagger(s)$ is unique.

$\square$

## C.2   PROOFS FOR SECTION 4.2

*Proof of Theorem 3.*

1. If $\xi_1, \xi_2 > 0$, then by Theorem 2, the user best response is always $q^*(i, s) = 0$ for all $i \in \{1, 2\}$ and $s \in [0, 1]$. We then compare the two separate cases of routing to model 1 and routing to model 2.

If the provider routes to model 2, their objective function value is $J_2(s, 0) = c_2\beta(0) = c_2/p_2$. On the other hand, if the provider routes to model 1, their objective function value is

$$J_1(s, 0) = \frac{c_1 + \frac{1-p_1}{p_2} c_2 s}{1 - (1 - p_1)(1 - s)}.$$

From Lemma 1, this function is monotone in $s$. To determine the direction, we take the first derivative

$$\frac{\partial J_1(s, 0)}{\partial s} = \frac{\frac{1-p_1}{p_2} c_2(1 - (1 - p_1)(1 - s)) - (1 - p_1)\left(c_1 + \frac{1-p_1}{p_2} c_2 s\right)}{(1 - (1 - p_1)(1 - s))^2}$$

$$= \frac{1 - p_1}{(1 - (1 - p_1)(1 - s))^2}\left(\frac{p_1}{p_2} c_2 - c_1\right)$$

First consider the case of $c_1/p_1 < c_2/p_2$. Here, we have $\partial J_1(s, 0)/\partial s > 0$, meaning that $J_1(s, 0)$ is monotone increasing and the optimal $s^* = 0$. Furthermore, $J_1(0, 0) = c_1/p_1 < c_2/p_2 < J_2(s, 0)$, meaning that the optimal policy is to route to model 1 without any escalation.

Conversely if $c_1/p_1 > c_2/p_2$, we have $\partial J_1(s, 0)/\partial s < 0$ meaning that $J_1(1, 0) \le J_1(s, 0)$ for all $s$. Furthermore, $J_1(1, 0) = c_1 + (1 - p_1)c_2/p_2 = c_1 + c_2/p_2 - p_1 c_2/p_2 > c_2/p_2 = J_2(s, 0)$ where the inequality falls from the condition on the cost-of-pass. Therefore, in this regime, the optimal policy is to route to model 2.

2. If $\xi_1, \xi_2 < 0$, then by Theorem 2, the user best response is always $q^*(i, s) = 1$ for all $i \in \{1, 2\}$ and $s \in [0, 1]$. We can compare the two separate cases of routing to model 1 versus routing to model 2.

   If the provider routes to model 2, their objective function value is $J_2(s, 1) = c_2 + P(1 - p_2)$. On the other hand, if the provider routes to model 1, their objective function value is $J_1(s, 1) = c_1 + P(1 - p_1)$, which is independent from $s$. Comparing these two objective function values yields

   $$J_1(s, 1) \le J_2(s, 1) \iff P \le \frac{c_2 - c_1}{p_2 - p_1}.$$

This completes the proof.

$\square$

*Proof of Theorem 4.* From Theorem 1, because $\xi_2 > 0$, the provider's return from routing to model 2 is $J_2(s, q^*) = c_2/p_2$. From Theorem 2, the provider's return from routing to model 1 is dependent on the escalation policy

$$J_1(s, q^*) = \begin{cases} c_1 + P(1 - p_1) & \text{if } s < s_0 \\ \dfrac{c_1 + \frac{1-p_1}{p_2} c_2 s}{p_1 + (1 - p_1)s} & \text{otherwise} \end{cases}$$

Furthermore from the proof to Theorem 3, $J_1(s, q)$ is monotone increasing if $c_1/p_1 < c_2/p_2$ and is monotone decreasing otherwise. Thus, for $s \ge s_0$, $J_1(s^*, q^*)$ is minimized at $s^* = s_0$ if the cost-of-pass of model 1 is lower than the cost-of-pass of model 2, and is minimized at $s^* = 1$ otherwise. Therefore, determining the provider-optimal policy simplifies to comparing $J_2(s, 0)$, $J_1(0, 1)$, and $\min\{J_1(s_0, 0), J_1(1, 0)\}$.

We first consider the case where $c_1/p_1 > c_2/p_2$. Here $J_1(1, 0) < J_1(s_0, 0)$ due to the monotone decreasing nature of the objective. Furthermore,

$$J_1(1, 0) = c_1 + \frac{1 - p_2}{p_1} c_2 = c_1 + \frac{c_2}{p_2} - \frac{p_1 c_2}{p_2} > \frac{c_2}{p_2} = J_2(s, 0)$$

meaning that routing to model 1 and escalating is strictly dominated by routing directly to model 2. Therefore, the optimal policy is to route to model 1 and set $s = 0$ if $c_1 + P(1 - p_1) < c_2/p_2$. Re-arranging this inequality yields the condition $P < P_1$. Conversely, if the condition is not satisfied, the optimal policy is to route to model 2.

We next consider the case where $c_1/p_1 < c_2/p_2$. Here, $J_1(s_0, 0) < J_1(1, 0)$ due to the monotone increasing nature of the objective. Furthermore

$$J_1(s_0, 0) - J_2(s, 0) = \frac{c_1 + \frac{1-p_1}{p_2} c_2 s_0}{p_1 + (1 - p_1)s_0} - \frac{c_2}{p_2} = \frac{c_1 - c_2}{p_1 + (1 - p_1)s_0} < 0$$

meaning that routing to model 2 is strictly dominated by first routing to model 1 and then escalating with probability $s_0$. Therefore, the optimal policy becomes to route to model 1 and set $s = 0$ if $c_1 + P(1 - p_1) < J_1(s_0, 0)$. Re-arranging this inequality yields the condition $P < P_2$. $\qquad\square$

*Proof of Theorem 5.* To determine the provider-optimal policy, we must check the objective function in four regimes: $J_1(s, 0)$ for $s \in [0, s_L]$, $J_1(s, q^\dagger(s))$ for $s \in (s_L, s_H)$, $J_1(1, 1)$, and $J_2(1)$. Note that $J_1(1, 1)$ subsumes $J_1(s, 1)$ for $s \in [s_H, 1]$, as the objective function is constant in this regime. For each of the three cases in the theorem statement, we prove optimality by checking against these regimes.

1. Note that $J_1(0, 0) = c_1/p_1$. We first show that $s = 0$ is optimal within the regime $[0, s_L]$ by proving that the objective function is monotone increasing in this regime:

$$\frac{\partial J_1(s, 0)}{\partial s} = \frac{1 - p_1}{(1 - (1 - p_1)(1 - s))^2} \left( \frac{p_1}{p_2} c_2 - c_1 \right) > 0$$

where the inequality arises from $c_1/p_1 < c_2/p_2$. Therefore, $J_1(0, 0) \leq J_1(s, 0) \forall s \in [0, s_L]$.

To show that $J_1(0, 0) \leq J_1(s, q^\dagger(s)) \forall s \in (s_L, s_H)$, we take the difference

$$J_1(s, q^\dagger(s)) - J_1(0, 0) \tag{10}$$

$$= C_1(s, q^\dagger(s)) - \frac{c_1}{p_1} + P\left(1 - S_1(s, q^\dagger(s))\right) \tag{11}$$

$$= C_1(s, q^\dagger(s)) - c_1 - \frac{c_1}{p_1}(1 - p_1) + P\left(1 - S_1(s, q^\dagger(s))\right) \tag{12}$$

$$= C_1(s, q^\dagger(s)) - c_1 - \frac{c_1}{p_1}\left(S_1(s, q^\dagger(s)) - p_1\right) + \left(1 - S_1(s, q^\dagger(s))\right)\left(P - \frac{c_1}{p_1}\right) \tag{13}$$

$$= \left(\frac{C_1(s, q^\dagger(s)) - c_1}{S_1(s, q^\dagger(s)) - p_1} - \frac{c_1}{p_1}\right)\left(S_1(s, q^\dagger(s)) - p_1\right) + \left(1 - S_1(s, q^\dagger(s))\right)\left(P - \frac{c_1}{p_1}\right) \tag{14}$$

$$\geq 0 \tag{15}$$

where (12) and (13) follow from adding and subtracting $c_1$ and $S_1(s, q^\dagger(s))$, respectively. Then, (14) follows from re-arranging the terms, and (15) follows from Lemma 6.

We then show that $s = 0$ yields lower cost than $s = 1$:

$$\frac{c_1}{p_1} < P \implies \frac{c_1}{p_1}(1 - p_1) < P(1 - p_1) \implies \frac{c_1}{p_1} < c_1 + P(1 - p_1) = J_1(1, 1)$$

Finally, to show that $(i, s) = (1, 0)$ yields lower cost than routing directly to model 2, we first observe the identity

$$\frac{c_1}{p_1} - \frac{c_1/p_1 - c_2}{1 - p_2} = \frac{c_2 p_1 - c_1 p_2}{p_1(1 - p_2)} \geq 0 \text{ if } \frac{c_2}{p_2} \geq \frac{c_1}{p_1}$$

Then,

$$P \geq \frac{c_1}{p_1} \geq \frac{c_1/p_1 - c_2}{1 - p_2} \implies \frac{c_1}{p_1} \leq c_2 + P(1 - p_2) = J_2(1)$$

2. Note that $J_1(1,1) = c_1 + P(1-p_2) = J_1(1,s)\forall s \in [s_H, 1]$. We first show that routing and escalating offers a lower objective function value than routing immediately to model 2:

$$P < \frac{c_2 - c_1}{p_2 - p_1} \implies c_1 + P(1-p_1) < c_2 + P(1-p_2) = J_2(1)$$

To show that $J_1(1,1) \le J_1(s, q^\dagger(s))\forall s \in (s_L, s_H)$, we take the difference

$$J_1(s, q^\dagger(s)) - J_1(1,1) = C_1(s, q^\dagger(s)) - c_1 - P\left(S_1(s, q^\dagger(s)) - p_1\right) \tag{16}$$

$$= \left(\frac{C_1(s, q^\dagger(s)) - c_1}{S_1(s, q^\dagger(s)) - p_1} - P\right)\left(S_1(s, q^\dagger(s)) - p_1\right) \tag{17}$$

$$\ge \left(\frac{c_1}{p_1} - P\right)\left(S_1(s, q^\dagger(s)) - p_1\right) \tag{18}$$

$$\ge 0 \tag{19}$$

where the second inequality follows from Lemma 6.

Finally, to show that $J_1(1,1) \le J_1(s,0)\forall s \in [0, s_L]$, we first note from Lemma 1 that $J_1(s,0) = (c_1 + (1-p_1)c_2 s/p_2)/(p_1 + (1-p_1)s)$ is monotone increasing when $c_2/p_2 > c_1/p_1$ and is monotone decreasing when $c_2/p_2 < c_1/p_1$. If this function is monotone decreasing, then

$$J_1(s,0) \ge J_1(0,0) = \frac{c_1}{p_1} > P \implies \frac{c_1}{p_1} > c_1 + P(1-p_1) = J_1(1,1).$$

On the other hand if $J_1(s,0)$ is increasing, then

$$J_1(s,0) - J_1(1,1) \ge J_1(1,0) - J_1(1,1) \tag{20}$$

$$= c_1 + \frac{1-p_1}{p_2}c_2 - c_1 - P(1-p_1) \tag{21}$$

$$= (1-p_1)\left(\frac{c_2}{p_2} - P\right) \tag{22}$$

$$\ge (1-p_1)\left(\frac{c_2}{p_2} - \frac{c_2 - c_1}{p_2 - p_1}\right) \tag{23}$$

$$= \frac{1-p_1}{p_2(p_2 - p_1)}(c_1 p_2 - c_2 p_1) \tag{24}$$

$$\ge 0 \tag{25}$$

where (23) follows from $P < (c_2 - c_1)/(p_2 - p_1)$, (24) re-arranges the terms, and (25) follows from the assumption that $c_1/p_1 > c_2/p_2$.

3. Note that $J_2(1) = c_2 + P(1-p_2)$. We first show that routing to model 2 immediately is more cost-effective than routing to model 1 and escalating on failure:

$$P < \frac{c_2 - c_1}{p_2 - p_1} \implies c_2 + P(1-p_2) < c_1 + P(1-p_1) = J_1(1,1).$$

To show that $J_2(1) \le J_1(s, q^\dagger(s))\forall s \in (s_L, s_H)$, we take the difference

$$J_1(s, q^\dagger(s)) - J_2(1) \tag{26}$$

$$= C_1(s, q^\dagger(s)) - c_2 - P\left(S_1(s, q^\dagger(s)) - p_2\right) \tag{27}$$

$$= C_1(s, q^\dagger(s)) - c_1 - (c_2 - c_1) - P\left(S_1(s, q^\dagger(s)) - p_1\right) + P(p_2 - p_1) \tag{28}$$

$$= \left(\frac{C_1(s, q^\dagger(s)) - c_1}{S_1(s, q^\dagger(s)) - p_1} - P\right)\left(S_1(s, q^\dagger(s)) - p_1\right) + P\left(p_2 - p_1 - (c_2 - c_1)\right) \tag{29}$$

$$\ge 0 \tag{30}$$

where the inequality follows from Lemma 6.

Finally, to show $J_2(1) \leq J_1(s, 0) \forall s \in [0, s_L]$, we note

$$P < \frac{c_1}{p_1} \implies c_1 + P(1 - p_1) < \frac{c_1}{p_1} = J_1(0, 0) \leq J_1(s, 0) \; \forall s \in [0, s_L]$$

where the final inequality follows from the monotone increasing nature of $J_1(s, 0)$.

$\square$

### C.3 PROOFS FOR SECTION 5

*Proof of Proposition 1.*

1. **If $\xi_1, \xi_2 > 0$:** then $q^*(i, s) = 0$ for all $(i, s)$ by Theorem 2. Moreover, $(i^*, s^*) = (\arg\min_i \{c_1/p_1, c_2/p_2\}, 0)$ from Theorem 3. To determine the user optimal routing strategy, we can compare

$$U_1(s, 0) - U_2(0) = \frac{\xi_1 + \frac{1-p_1}{p_2}\xi_2 s}{p_1 + (1 - p_1)s} - \frac{\xi_2}{p_2} = \frac{\frac{\xi_2 p_1}{p_2} - \xi_1}{p_1 + (1 - p_1)s}.$$

   If this difference is positive, then $(1, 0)$ is user-optimal, or otherwise $(2, 0)$. Thus, $\arg\max_{i,s} U_i(s, q^*(i, s)) = (2, 0)$ when $\xi_2/p_2 > \xi_1/p_1$, and is equal to $(1, 0)$ otherwise. Thus, $\Delta_U$ is equal to 0 if and only if $\xi_2/p_2 > \xi_1/p_1$ and $c_1/p_1 > c_2/p_2$ or if the signs of both inequalities are reversed.

2. **If $\xi_1, \xi_2 < 0$:** then $q^*(i, s) = 1$ for all $(i, s)$ by Theorem 2. Moreover, $(i^*, s^*) = (1, s)$ if $P \leq (c_2 - c_1)/(p_2 - p_1)$ and is equal to $(2, 0)$ otherwise. To determine the user optimal routing strategy, we compare

$$U_2(1) - U_1(s, 1) = \xi_2 - \xi_1,$$

   meaning that $\arg\max_{i,s} U_i(s, q^*(i, s) = (2, 0)$ when $\xi_2 > \xi_1$ and is equal to $(1, s)$ otherwise. Thus, $\Delta_U$ is equal to 0 if and only if $\xi_2 > \xi_1$ and $P < (c_2 - c_1)/(p_2 - p_1)$ or if the signs of both inequalities are reversed.

3. **If $\text{sign}(\xi_1) \neq \text{sign}(\xi_2)$:** We break this into two cases.

   (a) **If $\xi_1 < 0 < \xi_2$:** From Theorem 2, $q^*(2, s) = 0$ and $q^*(1, s) = \mathbb{1}\{s < s_0\}$. To determine the user optimal routing strategy, we can compare $U_2(0)$ versus $U_1(s, q^*(1, s))$. First, for any $s < s_0$:

$$U_2(0) - U_1(s, 1) = \frac{\xi_1 + \frac{1-p_1}{p_2}\xi_2}{p_1 + (1 - p_1)s} - \xi_1 = \frac{\xi_1(1 - p_1)(1 - s) + \frac{1-p_1}{p_2}\xi_2 s}{p_1 + (1 - p_1)s} > 0.$$

   Next for any $s \geq s_0$:

$$U_2(0) - U_1(s, 0) = \frac{\frac{\xi_2 p_1}{p_2} - \xi_1}{p_1 + (1 - p_1)s} > 0.$$

   Thus, $\Delta_U$ is equal to 0 if and only if $(i^*, s^*) = (2, 0)$.

   (b) **If $\xi_1 > 0 > \xi_2$:** From Theorem 2, $q^*(2, s) = 1$ and $q^*(1, s)$ is determined by three regimes. Starting from Lemma 3,

$$U_1(s, q) \leq U_1^-(\alpha) = \frac{\xi_1 + \xi_2 \alpha s}{1 - \alpha(1 - s)} \leq \frac{\xi_1}{p_1} = U_1(0, 0)$$

   where the second inequality arises from substituting $q = 0, s = 0$ into $U_1^-(\alpha)$. Therefore, the user prefers routing to model 1 without cascading among all choices that involve routing to model 1. Finally, we compare against model 2 to show $U_1(0, 0) - U_2(1) = \xi_1/p_1 - \xi_2 > 0$, meaning that the user utility is maximized overall by routing to model 1 and not escalating. Therefore, $\Delta_U = 0$ if and only if $(i^*, s^*) = (1, 0)$.

$\square$

### C.4 PROOFS FOR SECTION 6

*Proof of Proposition 2.* Before proving, we note that when $P \leq \min\{c_1/p_1, c_2/p_2\}$, we have the following two identities. First,

$$
\begin{aligned}
J_1(s, q) - J_1(0, 1) &= C_1(s, q) - c_1 - P(S_1(s, q) - p_1) \\
&= \left( \frac{C_1(s, q) - c_1}{S_1(s, q) - p_1} - P \right) (S_1(s, q) - p_1) \\
&\geq 0
\end{aligned}
$$

where the inequality rises from Lemma 6, showing $(C_1(s, q) - c_1)/(S_1(s, q) - p_1) \geq \min\{c_1/p_1, c_2/p_2\} \geq P$. Second,

$$
\begin{aligned}
J_2(q) - J_2(1) &= C_2(q) - c_2 - P(S_2(q) - p_2) \\
&= \left( \frac{C_2(q) - c_2}{S_2(q) - p_2} - P \right) (S_2(q) - p_2) \\
&= \left( \frac{c_2}{p_2} - P \right) (\beta(q) - 1) p_2 \\
&\geq 0
\end{aligned}
$$

where the inequality again rises from Lemma 6.

We now break the proof into two cases, each with two sub-cases.

1. **The effect of throttling model 1:**

   (a) **When $\xi_2 > 0$:** We must compare the effect on the provider-optimal cost between the settings $\xi_1, \xi_2 > 0$, where we let $J_{pre}^*$ be the optimal cost; and $\xi_2 > 0 > \xi_1$ where we let $J_{post}^*$ be the optimal cost. Then,

   $$
   \begin{aligned}
   J_{post}^* &\leq \min\{J_1(0, 1), J_1(s_0, 0), J_2(0)\} & (31) \\
   &= \min\{J_1(0, 1), J_2(0)\} & (32) \\
   &= \min\left\{ c_1 + P(1 - p_1), \frac{c_2}{p_2} \right\} & (33) \\
   &\leq \min\left\{ \frac{c_1}{p_1}, \frac{c_2}{p_2} \right\} & (34) \\
   &= J_{pre}^* & (35)
   \end{aligned}
   $$

   where (31) follows from Theorem 4, (32) follows from the identity stated at the start of the proof, (33) follows from substituting the the terms, and (34) follows from $P < c_1/p_1$. Finally, (35) follows from Theorem 3.

   (b) **When $\xi_2 < 0$:** We must compare the effect on the provider-optimal cost between the settings $\xi_1 > 0 > \xi_2$ and $0 > \xi_1, \xi_2$. Let $J_{pre}^*$ and $J_{post}^*$ be the provider-optimal cost in the first and second setting, respectively. Then,

   $$
   \begin{aligned}
   J_{pre}^* &= \min\{J_2(1), J_1(0, 0), J_1(s_L, 0), J_1(s, q^\dagger(s)), J_1(1, 1)\} & (36) \\
   &\geq \min\{J_2(1), J_1(0, 1)\} & (37) \\
   &= \min\left\{ c_2 + P(1 - p_2), c_1 + P(1 - p_1) \right\} & (38) \\
   &= J_{post}^* & (39)
   \end{aligned}
   $$

   where (36) follows from applying Theorem 2, (37) follows from the identity at the start of the proof, and (39) follows from Theorem 3.

2. **The effect of throttling model 2:**

   (a) **When $\xi_1 > 0$:** We must compare the effect of the provider-optimal cost between the settings $\xi_1, \xi_2 > 0$ where we let $J_{pre}^*$ be the optimal cost; and $\xi_1 > 0 > \xi_2$ where we let $J_{post}^*$ be the optimal cost. Then,

   $$
   J_{post}^* \leq \min\left\{ J_1(0, 0), J_2(1) \right\} \leq \min\{J_1(0, 0), J_2(0)\} = J_{pre}^*
   $$

where the first inequality follows from applying Theorem 2 and the second inequality follows from Theorem 3.

(b) **When $\xi_1 < 0$:** We must compare the effect of the provider-optimal cost between the settings $\xi_1 < 0 < \xi_2$ where we let $J^*_{pre}$ be the optimal cost; and $\xi_1, \xi_2 < 0$ where we let $J^*_{post}$ be the optimal cost. Then,

$$J^*_{pre} = \min\{J_1(0,1), J_1(s_0,0), J_2(0)\} \geq \min\{J_1(0,1), J_2(0)\}$$
$$\geq \min\{J_1(0,1), J_2(1)\} = J^*_{post}$$

where both inequalities draw on the identities presented at the start of the proof.

$\square$

