# OpenReview forum: "Routing, Cascades, and User Choice for LLMs"
_ICLR.cc/2026/Conference — ICLR 2026 Poster_

### Official Review · Reviewer_QacU · 2025-10-29

**Soundness:** 3
**Presentation:** 3
**Contribution:** 3
**Rating:** 8
**Confidence:** 3

**Summary:**

The paper studies how LLM providers route requests to different models (in this case, non-reasoning vs. reasoning) to balance performance, cost, and latency. It models this as a Stackelberg game between a cost-minimizing provider and a utility-maximizing user, who may re-prompt or abandon a task if it fails. The theoretical findings show that optimal routing is often a simple, static policy, but there could be a misalignment if the provider is incentivized to slow down responses in order to save costs.

**Strengths:**

- Instead of studying LLM routing in a vacuum, just from the perspective of the provider, it models the interaction as a Stackelberg game, which formally includes the reactive behavior of a user (who can re-prompt or abandon a task) in response to the provider's strategy.
- It shows a misalignment between provider and user when providers decide to slow down their models to nudge users to reduce their use.
- It derives simple and practical insights, like the fact that a static policy (routing to one model always) is better than cascading.
- The writing and motivation are clear

**Weaknesses:**

I don’t see major weaknesses, but perhaps my main concern is how relevant these findings and formulation will be long term. The paper deals with two models, one more capable and slower than the other. However, it seems like the direction from the main LLM providers is moving towards models with different levels of thinking capabilities depending on the effort. There, the model is routing in a way, and I’m not sure the findings here apply directly since the providers and users don’t have the same amount of control.

**Questions:**

- What do you think of your formulation in terms of models that automatically decide their effort based on each task?

---

> ### Author Response · Authors · 2025-11-17
> **Thanks for the detailed review**
>
> Thanks for your positive feedback on the novelty and clarity of the paper, as well as appreciating the key insights.
>
>
> > W1/Q1. What do you think of your formulation in terms of models that automatically decide their effort based on each task?
>
> This is a good point which we will mention in revision. Models that automatically decide their effort will still meet our major insights, but both the user and provider problems become more difficult.
>
> Currently, for a given task, each model $M_i$ has probability of success $p_i$. If models internally decide effort, then $p_i$ is a task-dependent variable (e.g., the model decides to put in a lot of effort, so $p_i$ is high). If $p_i$ is known to user and provider, then all our theorems stay the same.
>
> However, if the user does not know $p_i$, they must estimate it (or $\xi_i$) over multiple task instances via exploration/exploitation (see response to `pynF`). At the same time, the provider must estimate the user's belief distribution, using for example, observed user behavior in terminating chat sessions, thumbs-up/thumbs-down UI notifications, etc. Using the estimated beliefs, our results all hold.

---

> > ### Comment · Reviewer_QacU · 2025-11-23
> >
> > Thank you, that makes sense.
> >
> > This discussion reminded me of this recent NeurIPS 2025 paper [1]. The author also modeled the user-algorithm as a Stackelberg game and found that a costly signal (e.g., an option for the user to inform the algorithm's provider of an inconsistent preference) can reduce the burden of alignment, i.e., make it easier for the user to be understood.
> >
> > LLMs in mainstream providers now have these multiple levels of reasoning, and an option to abort the extended thinking. I wonder if that might be a possible addition, where the LLM takes into account that signal to reduce the burden for the user's utility.
> >
> > [1] Shirali, A. (2025). The Burden of Interactive Alignment with Inconsistent Preferences. arXiv preprint arXiv:2510.16368.

---

> > > ### Author Response · Authors · 2025-11-23
> > > **Thanks for sharing this reference.**
> > >
> > > Your suggestion is viable. At minimum for settings where user value is unknown, aborting "extended thinking" provides a  signal that can be used by the provider to estimate the user's internal $V$ or $\xi_2$. Further, an important next step is to consider the impact of user decisions, including i) giving the user choice to route; and ii) allow the user to abort or provide a signal if the autorouter behaves contrary to the user preference (see response to W1-iii for `M77r`).
> > >
> > > Note that this extension is only meaningful when users dynamically estimate model behavior & providers estimate user beliefs. In the full-information setting, the user will always force routes to their preferred model, making the extension trivial. We plan to explore giving users these actions in future work when studying the learning problem.

---

> > > > ### Comment · Reviewer_QacU · 2025-11-23
> > > >
> > > > Thanks again! I stand by my review, and I think this paper should be accepted.

---

### Official Review · Reviewer_pynF · 2025-10-29

**Soundness:** 2
**Presentation:** 2
**Contribution:** 3
**Rating:** 4
**Confidence:** 3

**Summary:**

This paper studies the interaction between a large language model (LLM) provider that routes tasks across multiple models and a user who can choose to reprompt or abandon tasks depending on model performance. The authors model this as a Stackelberg game between the provider and a user, where the provider decides a routing and cascading policy, and the user optimizes their abandonment policy based on perceived utility. The setup assumes two LLMs — a standard and a reasoning model — that differ in accuracy, latency, and cost. The paper provides a closed-form characterization of the equilibrium by first solving for the user’s best response (Theorems 1–2) and then deriving provider-optimal routing policies (Theorems 3–5). The results show that: (1) static routing without cascading is optimal in most regimes; (2) misalignment gaps arise when the provider’s cost-based model ranking differs from the user’s utility-based ranking (Section 5); provider throttling (i.e., intentionally increasing latency) can emerge as an equilibrium when user churn penalties are low (Proposition 2, Section 6). Empirically, the authors visualize user responses, provider policies, and misalignment gaps across parameter regimes (Figures 3–5) and provide intuitive insights on user patience, value-dominated vs. latency-dominated models, and welfare trade-offs.

**Strengths:**

Originality:The paper introduces a novel behavioral–economic perspective on LLM routing. While prior routing work (e.g., Chen et al., 2023; Ding et al., 2024; Hu et al., 2024) focuses on minimizing cost–latency trade-offs, this paper uniquely models strategic user response via a multi-round prompting game (Section 3). This Stackelberg formulation, with users as rational agents, represents a conceptual advance that bridges operations research and AI system design.

Quality: The analysis is mathematically sound. The closed-form results (Sections 4.1–4.2) are carefully derived and supported by additional lemmas in the appendix.

Clarity: The exposition is clear and well-structured. Visuals (Figures 1–5) effectively summarize equilibrium regions and threshold rules. The notation is consistent. The intuitive explanations following each theorem (especially Theorem 2 and 5) help readability.

Significance: This work is relevant for LLM system design and AI governance. The analysis gives explicit conditions for misalignment and welfare loss, which is useful for researchers and policymakers analyzing these AI platforms.

**Weaknesses:**

Empirical validation: The work is entirely theoretical. While this is appropriate for a conceptual contribution, the claims about user patience and latency manipulation (Figure 5 Right) would benefit from empirical support, e.g., simulations or user–provider experiments.

Limited model diversity: The analysis considers only two models (standard vs. reasoning). While the authors acknowledge this in the conclusion (Section 7), the extension to $n$ models could meaningfully affect equilibrium behavior—especially when users can select between several public endpoints.

Simplifying behavioral assumptions: The model assumes users observe provider cascade probabilities and adopt stationary abandonment policies (Section 3.1). In practice, users may have incomplete information or adaptive patience. A discussion of bounded rationality or stochastic user beliefs could improve this work.

**Questions:**

Extension to multiple models: How would the equilibrium generalize to $n>2$ models? Would the threshold structure persist or collapse into pairwise comparisons?

Dynamic user learning: The analysis assumes fixed $p_i$. How might the equilibrium change if users learn about model success probabilities over repeated interactions?

Alternative objectives: Have the authors considered a bi-level optimization where providers internalize user utility as part of a long-term revenue function?

---

> ### Author Response · Authors · 2025-11-17
> **Thanks for the detailed review**
>
> Thank you for your positive comments regarding the originality of the research, mathematical and writing quality, & practical significance.
>
> > W1. The work is entirely theoretical. While this is appropriate for a conceptual contribution, the claims about user patience and latency manipulation (Figure 5 Right) would benefit from empirical support, e.g., simulations or user–provider experiments.
>
> We agree that empirical evidence is important to this work. Our current figures use numerical simulations by calculating user & provider policies via Markov equations. We envision a more live experiment with user-providers as future work.
>
> > W2/Q1.  The analysis considers only two models (standard vs. reasoning). How would the equilibrium generalize to $n>2$ models?
>
> Thanks for this suggestion. Our framework can be extended by modifying the Markov chain with multiple model states and re-defining the absorption probabilities $S_i, L_i, C_i, P_i$. **We expect our key insights (i.e., limited opportunity for cascading; presence of misalignment when user-provider rankings disagree; potential for throttling) to still hold, but under more complex conditions.**
>
> Consider $n=3$. Assume $V p_i > t_i$ for all $i \leq n$. Following Theorem 1 and 2, if all models are value-dominated, users will  stay until success regardless of routing, i.e., $q^* = 0$. If the provider only routes without any cascading, success absorption is geometric and the optimal route is obtained by cost-of-pass $\arg\min_i {c_i/p_i}$. Furthermore, it must be optimal to select the most cost-efficient model right away rather than cascade to it.
>
> Unfortunately, closed-form analysis for $n>2$ becomes intractable. First, because each model can be value- or latency- dominated, there are $2^n$ scenarios to consider, and each scenario needs a separate user policy. Second, we can cascade from one model to multiple alternatives. Note that $s \in R^{n\times n}$ must be a matrix now. Nonetheless, using the machinery developed in Section 3, we can numerically optimize these policies.
>
> > W3. The model assumes users observe provider cascade probabilities and adopt stationary abandonment policies. A discussion of bounded rationality or stochastic user beliefs could improve this work.
>
> This is a great point and recommendation (see `d6sm` for related discussion). We agree that in practice, users are non-stationary, and they will not observe the cascade probability. Without full information, users must update an internal belief, but they can still make decisions with respected to an expected value $E[s]$. This extension preserves the key results, although it may present new opportunities for misalignment or user-suboptimal behavior. Our revision will also include a more detailed discussion on the impact of user patience according to bounded rationality when there is limited information.
>
> > Q2. The analysis assumes fixed $p_i$. How might the equilibrium change if users learn about model success probabilities over repeated interactions?
>
> Thanks for this suggestion. In this variation, users must act using a belief of $\xi_i$, which they can update after task instances.  This may permit a long-term exploration/exploitation strategy where users set $q$ to be small for early task instances to obtain better estimates of $p_i$ (and $\xi_i$). Theorem 1 and 2 can be applied with respect to $E[\xi_i]$.
>
> However, the provider problem becomes more difficult, because the provider policy depends on the user's belief of $\xi_i$, meaning that they must also estimate this (see respones to `M77r`). Our revision will include a discussion on this variant.
>
> > Q3. Have the authors considered a bi-level optimization where providers internalize user utility as part of a long-term revenue function?
>
> Thanks for this suggestion. The current setup is a bi-level optimization for the provider for a single task instance, where $P$ expresses the future potential lost revenue. Modeling long-term revenue would require a higher-order bi-level problem where providers and users must decide policy over multiple task instances. Here, users may control a probability of unsubscription $\pi(1-S(s, q))$, and decide: (i) when to abandon per-task; and (ii) when to unsubscribe over multiple instances. The provider problem becomes significantly more difficult to solve, but we still expect there to be potential for misalignment.

---

> > ### Comment · Reviewer_pynF · 2025-11-25
> >
> > Thank you for engaging in the rebuttal. I read your reply, as well as the chat with the other reviewers. My initial doubts were clarified, so I decided to raise my score.

---

> > > ### Author Response · Authors · 2025-11-29
> > >
> > > Thank you for raising your score!

---

### Official Review · Reviewer_d6sm · 2025-10-30

**Soundness:** 3
**Presentation:** 4
**Contribution:** 4
**Rating:** 8
**Confidence:** 2

**Summary:**

Advances in AI systems raise interesting questions for model deployers --- for any given user query, what model should be presented to the user, when models have different capabilities and costs to querying? And with a proliferation of model deployers, users too have a choice --- which deployer to engage with? The authors lay out these questions and formalize the interplay between model routing and user participation as a game, to which they develop game-theoretic insights into how actors may behave under different cost settings and how such a game could be "gamed" by model deployers in potentially harmful ways to users.

**Strengths:**

I very much enjoyed reading this paper! The authors' formalization of the problem of user and model deployer interaction as it relates to what model is served is a highly (and increasingly) important to today's consumer AI dynamic. I found the connection to game theory quite creative, and I learned quite a bit from reading the paper. The authors' novel insights about possible gameification that model deployers could engage in, given such a game (re throttling latency) are likely quite a valuable contribution in their own right. I appreciated that the authors were open about their limitations of their set-up of the problem as well. I imagine there could be a nice blossoming literature around this and related works to better understand and model people and deployers' choices in the setting of multi-model choice.

**Weaknesses:**

I found quite few weaknesses in the work; however, I may have missed something in the mathematics. As someone a bit weaker on the theory-side, I did find some of the theoretical discourse quite dense and a little convoluted -- especially section 3 (but this may be my own naivete --- indicated in my lower confidence score).

As noted above, the authors seem quite upfront on their limitations (I would be interested in settings where the user may not be aware of the routing policy or s).

More minor but important -- I found some of the visuals a little confusing. There are quite a few colors in Fig 2 but it is not clear what they relate to. It would be helpful if the caption spelled out the assignment of colors to differences in the kind of computation. Figure 3 also indicates that Model 1 is shown with dashed lines, but I can't seem to see where this is?

**Questions:**

See above re: questions on interpreting figure 4 (where is the Model 1 dashed line?)

I would be interested in the authors' extension to not just other models -- but the case where multiple model deployers are simultaneously competing for the same user.

---

> ### Author Response · Authors · 2025-11-17
> **Thanks for the detailed review**
>
> Thank you very much for sharing your positive experience with the paper!
>
> > W1. I did find some of the theoretical discourse quite dense and a little convoluted -- especially section 3
>
> We appreciate the concern. The formalism in Section 3 was intended to ensure that the key components of our problem ($C_i, S_i, L_i$) are well-justified. We will add more exposition in Section 3 if given an additional page in the camera-ready version. Specifically, we will give more intuition of the user and provider objectives (see the motivating example in response to reviewer `M77r`), and a slower walk-through of the transition logic and corresponding derivations of the main components.
>
>
> > W2. As noted above, the authors seem quite upfront on their limitations (I would be interested in settings where the user may not be aware of the routing policy or s).
>
> Thanks for your suggestion of this important extension. Our work is partially motivated by ChatGPT 5.0, where the web interface indicates to which model has been routed (but not $s$). For LLM providers that do not provide the routing policy $s$, the user can still optimize by using a belief distribution of $s$ estimated by observing prior routes. Given the belief distribution, the user can compare $E[s]$ against $s_0, s_L, s_H$ (see response to reviewer `pynF`). The user results (Theorem 1 and 2) hold but using $E[s]$, and the provider results remain unchanged.
>
> > W3/Q1. I found some of the visuals a little confusing. questions on interpreting figure 4 (where is the Model 1 dashed line?)
>
> Thanks for this suggestion. The Figure 4 legends are slightly erroneous. The hatch pattern in the legend should be both hatched and shaded blue. That is, Model 1 ($s=0.04$) is the top-left in Figure 4 (Left) and Model 1 ($s=1$) is the bottom in Figure 4 (Right).
>
> > Q2. I would be interested in the authors' extension to not just other models -- but the case where multiple model deployers are simultaneously competing for the same user.
>
> This is a great question. We currently treat the effect of competition implicitly via the penalty $P$, which measures the expected risk of losing future revenue if users abandon tasks. If there are multiple competing providers, users are more likely to unsubscribe, meaning $P$ is a function of the competition (i.e., the price and performance of all providers). This preserves and strengthens our current insights, e.g., if $P$ is large, then Theorem 3 & 4 imply we need to route to M2 more, and Proposition 2 implies we have less chances of throttling.
>
> To fully model the effect of competition would require optimizing both routing policies and subscription prices [1, 2]. This could lead to a second-order complex game where the model provider must compete with other providers on price and quality to satisfy the user. Since this requires significant additional complexity, we leave it to future work.
>
>
> [1] https://arxiv.org/abs/2502.07736
>
> [2] https://arxiv.org/abs/2411.02661

---

> > ### Comment · Reviewer_d6sm · 2025-11-24
> > **Strong paper**
> >
> > I agree with reviewer QacU. I appreciate the depth of responses and stand by endorsing the paper's acceptance!

---

### Official Review · Reviewer_M77r · 2025-10-31

**Soundness:** 2
**Presentation:** 2
**Contribution:** 1
**Rating:** 2
**Confidence:** 3

**Summary:**

The paper’s goal is to characterize operation regime of LLMs where providers offer models of different capability (and hence difference prices) and the users aim at minimizing their costs while getting the best utility out of the model. Some other variables of interest are the provider’s desire to minimize their inference costs and the user’s desire to minimize the latency. The paper studies multiple strategies, namely, routing to the best possible model and cascading through small models first and then sending to the larger model if the smaller model’s response is not good enough. The paper then proves different statements about optimal routing / cascading policies given different capability and latency ratios between the two models.

**Strengths:**

1. The high level problem of learning optimal routing strategies is quite well motivated.
2. The formalism is thorough, though it could be quite dense at some points.

**Weaknesses:**

1. In general, I have quite a few concerns about how realistic the whole setup is. The tasks that the paper defines seem a bit different from what LLM users encounter in practice. The paper should describe (i) Why is the monetary cost to the user is not modeled? (ii) The user churning with some predefined probability seems plausible. But during deployment, users mostly cannot check the accuracy on each individual example. So is the framework here meant to be more suitable for scenarios when users are in the model testing phase? (iii) Why would the users keep trying the same model M1 if it doesn’t respond correctly? (iv) Users generally test on a set of independent queries. If the performance is bad on multiple successive queries, users could abandon the provider. How are the dynamics over multiple queries modeled? (v) Why is $s$ a number between 0 and 1. Shouldn’t it always be 1? If M1 fails, shouldn’t the provider always cascade to M2? (vi) The cascading step seems to assume that the model provider knows if the model answered the prompt correctly. Why is this a realistic assumption? In reality, the model providers are not even supposed to see the user inputs due to privacy and IP reasons. (vii) The notion of “value” that the user derives from the model is quite vague. Is this a real number or simply 0/1 accuracy. If it is the former, why?

2. On a similar note, in Section 6, if the service costs are so high, why can the provider simply not raise the price rather than throttling users (disgruntled users might never return)? Not clear why this would be a rational policy.

3. The value of the dense formalism is not clear and the insights that we draw looks to be derivable from simpler analyses. The main idea seems to be to compare the ratio between the accuracy and latency. The paper correctly points out that when the accuracies are the same, cascading doesn’t make sense since it add unnecessary latency by putting the smaller model in front of the bigger model. The results seem quite intuitive when considering this tradeoff. For instance, about the insight in Line 88, of course we expect users to stay if the net value is positive. The difficulty in practice is that it is very difficult to judge in advance for an unseen data point if the model will provide positive values, e.g., of the summary of an article will be good enough. Similarly, the insight in line 85 also seems straightforward. The main trouble is that we cannot predict when one model provides better value than the other (see weakness 1).

**Questions:**

Please see questions under weakness 1.

**Details Of Ethics Concerns:**

Reading the Appendix A and text in line 476, it seems that LLMs contributed very substantially to the ideation of the paper. For instance, the authors not that LLMs were used in "(i) deriving proof steps or searching for counterexamples; (ii) preparing first drafts for sections and editing with feedback; and (iii) coding plots.". For proofs, the process involved 2-10 prompts (Table 1).

**Given that LLMs seem to have made fundamental contributions (to the extent to be considered authors), I would like to flag the paper for adherence to the ICLR policies.** The guidelines on the website do not clearly state what extent of LLM involvement is permitted.

---

> ### Author Response · Authors · 2025-11-17
> **Thanks for the detailed review (1/2)**
>
> Thanks for your positive feedback on the thoroughness of our framework.
>
> > W1. Realism of the setup
>
> We are studying the scenario where a user pays a subscription price to allow unlimited uses of an LLM. To better motivate our work, consider the following example:
>
>
> >> A user subscribed to an LLM provider asks the LLM to prove a theorem, create a presentation, etc. The provider routes the user’s task to either a standard or reasoning model (e.g., GPT-5 standard/thinking). The user reviews the LLM output (i.e., verifies the proof, reviews the slides). Completing the task gives the user value (a paper is completed, billable work-hours are saved, etc.), while using the LLM costs the user time. If the user is unsatisfied with the output, they prompt again, after which the LLM must again decide where to route. If the user is repeatedly unsatisfied, they give up using the LLM and decide to do the task themself. If this happens many times, the user may eventually cancel the subscription, causing monetary penalty to the provider.
>
>    We study a stylized version of this example where rewards/costs/transition probabilities are in steady-state. These assumptions allow for interpretion of  the user-provider interaction dynamics. **Our revision will include the above motivating example and the below comments to improve clarity.**
>
> > W1.i) Why is the monetary cost to the user not modeled?
>
> In the subscription, the user pays a fixed fee up front and faces only latency and quality trade-offs at interaction time. The monetary cost to the user is constant and does not affect per-task routing decisions. In API use, we might define $V p_i - t_i - s_i$ where $s_i$ is a per-prompt cost.
>
> > W1.ii) Users can’t verify correctness per example
>
> We are motivated by in-the-wild usage where users verify success/correctness in terms of satisfaction (e.g., verifying correctness of proof steps, reviewing slides until they are satisfied). This satisfaction is observed when the user clicks "thumbs up" on the UI, thanks the chatbot, or decides the response is good enough to stop, while failure involves the user continuing the conversation with additional requests.
>
> > W1.iii) Why would the users keep trying the same model M1
>
> The provider chooses the route and cascade probability [1], while the user only chooses whether to abandon. From your suggestion, we agree that an important next step is to study user-controls or user-overrides on the route. Although this requires more formalism, we expect the core insights (e.g., cascading being sub-optimal, the existence of misalignment) to still be driven by a user net-value vs provider cost-of-pass comparison.
>
> > W1.iv) Users generally test on a set of independent queries. If the performance is bad on multiple successive queries, users could abandon the provider. How are the dynamics over multiple queries modeled?
>
> Currently, the effects of multiple independent tasks are approximated in the penalty $P$, i.e., the expected loss in revenue from a future potential unsubscription conditioned on the user abandoning the task.
>
> We could extend our work to multiple independent tasks by treating $P_i$ as dependent on the $i$-th task (see response to reviewer `pynF`). For example, $P_i$ may grow with each failure and be a higher-level user decision variable. **Our results will hold for each fixed task instance**, but we leave the corresponding higher-order problem to future work.
>
> > W1.v) Why is $s \in [0,1]$ rather than always $s=1$?
>
> We permit a probabilistic routing policy [1]. Always cascading is not optimal due to the probabilistic nature of user inputs and LLM outputs (see Theorem 3-5). For instance, if a user fails to include enough context in their first prompt, the standard model may be incorrect, but re-prompting it with more context may suffice.
>
> > W1.vi) The cascading step assumes that the model provider knows if the model answered the prompt correctly.
>
> We assume that if the user is continuing to prompt in the LLM session, the task has not been completed; if the user stopped, either the LLM was successful or the user abandoned. Most importantly, **the provider does not need to know if they were successful, just that the user is continuing the session.**
>
> > W1.vii) The notion of "value" that the user derives is vague. ?
>
> We model $V$ as a real number that reflects how much the user cares about a task (e.g., time saved from using the LLM). This abstract quantity allows us to quantify user utility $\xi_i$ via a standard linear econometric model.
>
> **Crucially, the provider does not need to know V; they only need to estimate if $\xi_i > 0$,** for example by observing user behavior in terminating chat sessions, thumbs-up/thumbs-down UI notifications, etc. We envision potential future work on estimating $\xi_i$ and studying the impact of estimation error on routing outcomes for users.
>
>
> [1] https://openreview.net/pdf?id=AAl89VNNy1

---

> > ### Author Response · Authors · 2025-11-17
> > **Thanks for the detailed review (2/2)**
> >
> > > W2. If the service costs are so high, why can the provider simply not raise the price rather than throttling users?
> >
> > This is a great question and we will clarify in the revision. We explore routing/throttling behavior given a fixed subscription price. Raising prices may not always be viable due to competition; a provider with twice the internal costs as their competitor cannot charge twice as much to subscribers. Throttling is a potential hidden price increase that can reduces provider costs.
> >
> > > W3. The value of the formalism is not clear and the insights looks to be derivable from simpler analyses. The main trouble is that we cannot predict when one model provides better value than the other.
> >
> > We develop foundations to analyze the user behavior interactions of routing, which is unexplored in the routing literature. While some high-level results may seem intuitive, the interactions (e.g., regions when cascades are optimal or when there is mis-alignment) are non-trivial. The key difficulty comes when models are differentiated in their accuracy latency trade-offs $\xi_i$ (i.e., Theorem 2, 4, 5). Moreover, we note from reviewer `d6sm` on the novelty and significance of revealing gamification via throttling. Finally, we emphasize that providers do not need to know $V$, they only have to estimate when $\xi_i > 0$.
> >
> > > W4. Ethics Concerns.
> >
> > We appreciate the concern and take it seriously. **All work was thoroughly revised by the authors. We alone accept full responsibility for the correctness of the work. We do not consider an LLM to be an author.**  LLMs were used as assistive tools, analogous to a theorem-proving software or a writing aid. For instance, an LLM can derive the steps to a proof, but these steps are manually verified for correctness and revised for ease of interpretation. We disclose all LLM usage transparently, following the ICLR 2026 disclosure policies. To the best of our knowledge, our usage does not violate ICLR 2026 policy.

---

### Author Response · Authors · 2025-11-18
**Summary: Thanks for all reviews**

We thank all reviewers for their constructive and positive feedback:

- Well-motivated and novel formulation of LLM routing with user response (`M77r, d6sm, pynF, QacU`)
- Thoroughness of formalism and mathematical results (`M77r, pynF`)
- Practical insights drawn from the framework (`d6sm, pynF, QacU`)
- Quality & clarity of writing (`d6sm, pynF, QacU`)


Our paper introduces the novel problem of the interaction between routed LLMs and users.  We build a stylized model of a prompting game between user and provider. This allows us to derive closed-form user/provider policies and identify the potential for misalignment between user and provider objectives. This is a fundamental first step towards understanding more empirical work on practicalities of LLM routing.


We received several common questions. We will revise the paper with the corresponding discussions:

- **Partial information extension:** Our framework assumes the user knows $p_i$ and $s$. If $p_i$ is not available, users can still use a belief distribution of $\xi_i(p_i)$ to estimate $E[\xi_i]$ and correspondingly apply Theorem 1 and 2. This extension poses two next steps: (i) the provider optimal policy requires estimating the user's belief of $E[\xi_i]$; and (ii) users may try an exploration/exploitation strategy over multiple task instances where they set low $q$ at first to estimate $\xi_i$ with more observations. We still expect to observe potential misalignment.
- **Extensions to multiple models:** The framework generalizes to $n>2$ models by modifying the Markov chain, re-defining absorption probabilities for $S_i, L_i, C_i$, and numerically optimizing the user/provider problems. We expect the key insights of our work to hold. However, there are $2^n$ scenarios because the user optimal response depends on which combination of models satisfy $\xi_i > 0$. Consequently, we do not expect neat closed-form solutions for all settings.
- **Realism of model and formal setup:** Our stylized model tries to give the most generalization while permitting closed-form theoretic results. We will include a more expanded motivating example to explain our design choices.

---

### Author Response · Authors · 2025-12-03
**Final Summary to ACs, SACs, PCs**

Dear ACs, SACs, PCs,

Thank you for your consideration of our work and your efforts in reviewing this paper after the shuffle. We are writing to summarize our review experience.

This paper received on average positive reviews, but with high variance. We refer to the summary comment below for details. Feedback from all reviewers were insightful, but primarily consisted of clarifying questions regarding the setup of our model or additional extensions. In our rebuttals, we have clarified all the key points. We plan to include the below discussion points using the extra page limit in the final version of the paper if it is accepted.

In their initial review, `d6sm` and `QacU` scored the paper 8. After the rebuttal, they both independently posted in favor of acceptance of the paper. In their initial review, `pynF` scored the paper 4. After our rebuttal, they increased their score to 8 and posted positively in the comments. This has unfortunately been reset. In their initial review, `M77r` scored the paper 2, but unfortunately, they were not able to respond to our rebuttal before the deadline. We hope that we have sufficiently clarified the model concerns of `M77r` in our rebuttal.

We hope this context helps you better interpret the current scores in light of the full discussion record.

---

### Meta-Review · Area_Chair_Eciv · 2025-12-11

**Summary:**

Strengths:

i) Clear motivation for routing and cascading.

ii) Well-defined mathematical formulation.

iii) A new perspective on routing and cascading in LLM agents, and novel insights.

Main Concerns:

i) The problem setting remains somewhat simplified. Aspects such as multiple models, model pricing, parameter estimation, and adaptive reasoning level control are not incorporated. Including these elements would make the framework more realistic and more aligned with practical LLM-agent systems.

ii) The analysis is relatively simple, since the underlying model is a toy example.

**Reviewer Concerns:**

In the rebuttal, the authors provide extensive discussion on how the model could be extended to better align with real systems. This partially addresses the concern regarding realism. More importantly, I believe the key contribution of this paper lies in offering a novel perspective on routing and cascading in LLM agents. This perspective has the potential to stimulate a new line of research, where subsequent work can progressively incorporate more realistic components into the framework. For this reason, I do not view the simplified model and analysis as a critical flaw.

**Reviewer Scores:**

Reviewer M77r considers the problem setting to be far from realistic, and the authors provide substantial discussion on this point in the rebuttal. I believe that, after the discussion phase, he may increase his score by 1, or possibly 2.

Reviewer d6sm and QacU will not change the scores.

Reviewer pynF's initial doubts were clarified, so it is possible that he may raise his score by up to 2 points.

---

### Decision · Program_Chairs · 2026-01-26

Accept (Poster)